# Loss of CTRP10 results in female obesity with preserved metabolic health

**Fangluo Chen[1,2], Dylan C Sarver[1,2], Muzna Saqib[1,2], Leandro M Velez[3,4], Susan Aja[2,5], Marcus M Seldin[3,4], G William Wong[1,2]***

[1]Department of Physiology, Johns Hopkins University School of Medicine, Baltimore, United States; [2]Center for Metabolism and Obesity Research, Johns Hopkins University School of Medicine, Baltimore, United States; [3]Center for Epigenetics and Metabolism, University of California, Irvine, Irvine, United States; [4]Department of Biological Chemistry, University of California, Irvine, Irvine, United States; [5]Department of Neuroscience, Johns Hopkins University School of Medicine, Baltimore, United States

## eLife Assessment

This manuscript presents a detailed characterization of male and female wildtype and Ctrp10 knockout mice, and reveals that knockout mice develop female-specific obesity that is largely uncoupled from metabolic dysfunction. The data are **convincing**, and the work will be an **important** contribution to understanding how obesity is coupled to metabolic dysfunction, and how this can occur in a sex-specific manner.

*For correspondence:
gwwong@jhmi.edu

**Competing interest:** The authors declare that no competing interests exist.

**Abstract** Obesity is a major risk factor for type 2 diabetes, dyslipidemia, cardiovascular disease, and hypertension. Intriguingly, there is a subset of metabolically healthy obese (MHO) individuals who are seemingly able to maintain a healthy metabolic profile free of metabolic syndrome. The molecular underpinnings of MHO, however, are not well understood. Here, we report that CTRP10/ C1QL2-deficient mice represent a unique female model of MHO. CTRP10 modulates weight gain in a striking and sexually dimorphic manner. Female, but not male, mice lacking CTRP10 develop obesity with age on a low-fat diet while maintaining an otherwise healthy metabolic profile. When fed an obesogenic diet, female *Ctrp10* knockout (KO) mice show rapid weight gain. Despite pronounced obesity, *Ctrp10* KO female mice do not develop steatosis, dyslipidemia, glucose intolerance, insulin resistance, oxidative stress, or low-grade inflammation. Obesity is largely uncoupled from metabolic dysregulation in female KO mice. Multi-tissue transcriptomic analyses highlighted gene expression changes and pathways associated with insulin-sensitive obesity. Transcriptional correlation of the differentially expressed gene (DEG) orthologs in humans also shows sex differences in gene connectivity within and across metabolic tissues, underscoring the conserved sex-dependent function of CTRP10. Collectively, our findings suggest that CTRP10 negatively regulates body weight in females, and that loss of CTRP10 results in benign obesity with largely preserved insulin sensitivity and metabolic health. This female MHO mouse model is valuable for understanding sex-biased mechanisms that uncouple obesity from metabolic dysfunction.

## Introduction

The prevalence of obesity has nearly tripled in the past four decades and the underlying cause is complex and multifactorial (***NCD Risk Factor Collaboration (NCD-RisC), 2017***; ***Flier, 2023***). Genetics, environmental and social factors, and demographics all play a role in contributing to excessive weight

gain in the setting of overnutrition (*Blüher, 2019*; *Bouchard, 2021*). Although obesity is a major risk factor for type 2 diabetes, dyslipidemia, cardiovascular disease, and hypertension, not all obese individuals develop the metabolic syndrome (*Loos and Kilpeläinen, 2018*). There is a subset of metabolically healthy obese (MHO) individuals with an apparently healthy metabolic profile free of some or most components of the metabolic syndrome (*van Vliet-Ostaptchouk et al., 2014*; *Blüher, 2020*). The molecular and physiological underpinnings of MHO are, however, not well understood. Novel preclinical animal models that can recapitulate features of MHO will be valuable in illuminating pathways that resist the deleterious effects of obesity and provide new therapeutic avenues to mitigate obesity-linked comorbidities.

The mechanisms that normally maintain body weight and metabolic homeostasis are complex and involve both cell autonomous and non-cell autonomous mechanisms. Tissue crosstalk mediated by paracrine and endocrine factors plays an especially important role in coordinating metabolic processes across organ systems to maintain energy balance (*Priest and Tontonoz, 2019*). Of the secretory proteins that circulate in plasma, C1q/TNF-related proteins (CTRP1-15) have emerged as important regulators of insulin sensitivity, and glucose and lipid metabolism (*Seldin et al., 2014*). We originally identified the first seven members of the CTRP family based on shared sequence homology to the insulin-sensitizing adipokine, adiponectin (*Wong et al., 2004*), and subsequently characterized eight additional members (*Seldin et al., 2012*; *Wei et al., 2012*; *Wei et al., 2011*; *Wei et al., 2013*; *Wong et al., 2009*; *Wong et al., 2008*). All 15 CTRPs share a common C-terminal globular C1q domain and are part of the much larger C1q family (*Ghai et al., 2007*; *Ressl et al., 2015*). The use of gain- and loss-of-function mouse models has helped establish CTRP's role in controlling various aspects of sugar and fat metabolism (*Wei et al., 2012*; *Lei et al., 2016*; *Lei et al., 2017*; *Lei and Wong, 2019*; *Little et al., 2019*; *Petersen et al., 2017*; *Rodriguez et al., 2016*; *Sarver et al., 2020*; *Sarver et al., 2022a*; *Sarver et al., 2022b*; *Tan et al., 2020*; *Tan et al., 2016*; *Wei et al., 2014*; *Wolf et al., 2016*; *Peterson et al., 2014*; *Peterson et al., 2013a*; *Peterson et al., 2013b*). Additional diverse functions of CTRPs have also been demonstrated in the cardiovascular (*Appari et al., 2017*; *Zheng et al., 2011*; *Kambara et al., 2012*; *Kambara et al., 2015*; *Kanemura et al., 2017*; *Ogawa et al., 2020*; *Otaka et al., 2018*; *Uemura et al., 2013*; *Yuasa et al., 2016*; *Sun et al., 2013b*; *Yi et al., 2012*; *Han et al., 2018*; *Lee et al., 2022*), renal (*Rodriguez et al., 2021*; *Rodriguez et al., 2020*), immune (*Lei et al., 2017*; *Kirketerp-Møller et al., 2020*; *Lahav et al., 2021*), sensory (*Ayyagari et al., 2005*; *Hayward et al., 2003*), gastrointestinal (*Luo et al., 2016*), musculoskeletal (*Youngstrom et al., 2020*; *Hamoud et al., 2018*; *Cho et al., 2021*), and the nervous system (*Kakegawa et al., 2015*; *Sigoillot et al., 2015*; *Martinelli et al., 2016*; *Sarver et al., 2021*).

Of the family members, CTRP10 (also known as C1QL2) is understudied and consequently only limited information is available concerning its function. The best characterized role of CTRP10 is in the central nervous system (CNS). It has been shown that CTRP10 secreted from mossy fibers is required for the proper clustering of kainite-type glutamate receptors on postsynaptic CA3 pyramidal neurons in the hippocampus (*Matsuda et al., 2016*). It serves as a transsynaptic organizer by directly binding to Neurexin3 (Nrxn3) on the presynaptic terminals, and to glutamate ionotropic receptor kainate type subunit 2 (GluK2) and GluK4 on the postsynaptic terminals (*Matsuda et al., 2016*). Additional putative roles of CTRP10 in the CNS have also been suggested. Genome-wide association studies (GWAS) have implicated CTRP10/C1QL2 in cocaine use disorder (*Huggett and Stallings, 2020*). In rat models of depression, *Ctrp10* expression is increased in the dentate gyrus and reduced in the nucleus accumbens (*Unroe et al., 2021*). In humans with a history of psychiatric disorders (e.g. schizophrenia), the expression of *CTRP10* is elevated in the dorsolateral prefrontal cortex of both males and females (*Unroe et al., 2021*). Whether and how CTRP10 contributes to addictive behavior and psychiatric disorders is unknown. Recent large-scale efforts to illuminate the genetic architecture of the human plasma proteome also highlighted an association of plasma C1QL2/CTRP10 levels with a missense C4BPA variant (*Sun et al., 2023*). In pregnancy, circulating C1QL2/CTRP10 levels are negatively associated with the plasma inflammasome marker, α1-acid glycoprotein (AGP) (*Kim et al., 2024*). The biological significance of these clinical observations, however, remains unclear.

The potential function of CTRP10 in peripheral tissues, however, is essentially unknown and unexplored. The present study was motivated by the well-documented metabolic functions of many CTRP family members we have characterized to date using genetic loss-of-function mouse models (*Lei et al., 2016*; *Lei et al., 2017*; *Lei and Wong, 2019*; *Little et al., 2019*; *Petersen et al., 2017*;

*Rodriguez et al., 2016*; *Sarver et al., 2020*; *Sarver et al., 2022a*; *Sarver et al., 2022b*; *Tan et al., 2020*; *Wei et al., 2014*; *Wolf et al., 2016*). We determined that the expression of *Ctrp10* in peripheral tissues is modulated by diet and nutritional states, and thus may have a metabolic role. We therefore used a genetic loss-of-function mouse model to determine if CTRP10 is required for regulating systemic metabolism. We unexpectedly discovered a female-specific requirement of CTRP10 for body weight control. We showed that the *Ctrp10* KO mice represent a unique female model of MHO with largely preserved insulin sensitivity and metabolic health. This valuable mouse model can be used to inform sex-dependent mechanisms that uncouple obesity from insulin resistance, dyslipidemia, and metabolic dysfunction.

## Results

### Nutritional regulation of *Ctrp10* expression in the brain and peripheral tissues

CTRP10 protein is highly conserved from zebrafish to human (*Figure 1A*), with an amino acid identity of 67%, 71%, 77%, and 94% between the full-length human protein and the fish, frog, chicken, and mouse orthologs, respectively. The conservation is much higher at the C-terminal globular C1q domain (93–100% identity) between the orthologs. Among the 12 different mouse tissues examined, the brain had the highest expression of *Ctrp10* (*Figure 1B*), consistent with previous findings (*Iijima et al., 2010*). Expression of *Ctrp10* in most peripheral tissues was lower than the brain (*Figure 1B*). We first determined whether *Ctrp10* expression is modulated by nutrition and metabolic state. Male mice (~10 weeks old) were subjected to fasting and refeeding. In the refed period after an overnight fast, we observed a significant downregulation of *Ctrp10* in the visceral (gonadal) white adipose tissue (gWAT), liver, skeletal muscle, kidney, cerebellum, cortex, and hypothalamus relative to the fasted state (*Figure 1C*). Next, we examined whether an obesogenic diet alters the expression of *Ctrp10*. Male mice fed a high-fat diet (HFD) for 12 weeks had a modest increase in *Ctrp10* expression in brown adipose tissue (BAT) and heart and decreased expression in skeletal muscle relative to mice fed a control low-fat diet (LFD; *Figure 1D*). These data indicate that *Ctrp10* expression is dynamically regulated by acute alterations in energy balance, and perhaps to a lesser extent to chronic changes in nutritional state on an obesogenic diet.

### Generation of *Ctrp10* knockout (KO) mice

We used mice lacking CTRP10 to address whether this secreted protein has a metabolic role in vivo. The mouse *Ctrp10* gene consists of two exons (*Figure 1E*). The CRISPR-Cas9 method was used to remove the entire protein coding region spanning exon 1 and 2, thus ensuring a complete null allele (*Figure 1E–F*). The targeted allele was confirmed by sequencing. As expected, based on the gene deletion strategy, the *Ctrp10* transcript was absent from KO mice (*Figure 1G*).

### CTRP10 is largely dispensable for metabolic homeostasis in young mice fed a control LFD

We first addressed the requirement of CTRP10 in maintaining baseline metabolic homeostasis by assessing metabolic parameters of mice fed a LFD. The body weight and body composition of male mice fed a LFD were not different between genotypes (*Figure 2A–B*). By 20 weeks of age, female KO mice fed LFD had a modestly higher body weight relative to WT controls (*Figure 2C*), although the body composition was not different between genotypes (*Figure 2D*). Food intake, physical activity, and energy expenditure as measured by indirect calorimetry were also not different between genotypes of either sex across the circadian cycle (light and dark) and metabolic states (ad libitum fed, fasted, refed; *Figure 2E–J*). Because *Ctrp10* expression is regulated by nutritional states (*Figure 1C*), we assessed serum metabolite levels in WT and KO mice in response to fasting and refeeding. No significant differences in fasting and refeeding blood glucose, serum insulin, triglyceride (TG), cholesterol, non-esterified free fatty acids (NEFA), and β-hydroxybutyrate levels were observed between genotypes of either sex, except the female KO mice had slightly lower fasting β-hydroxybutyrate levels (*Figure 3A–B*). We performed glucose and insulin tolerance tests to determine any potential differences in glucose handling capacity and insulin sensitivity. No significant differences in glucose and insulin tolerance were noted between genotypes of either sex (*Figure 3C–F*). Together, these

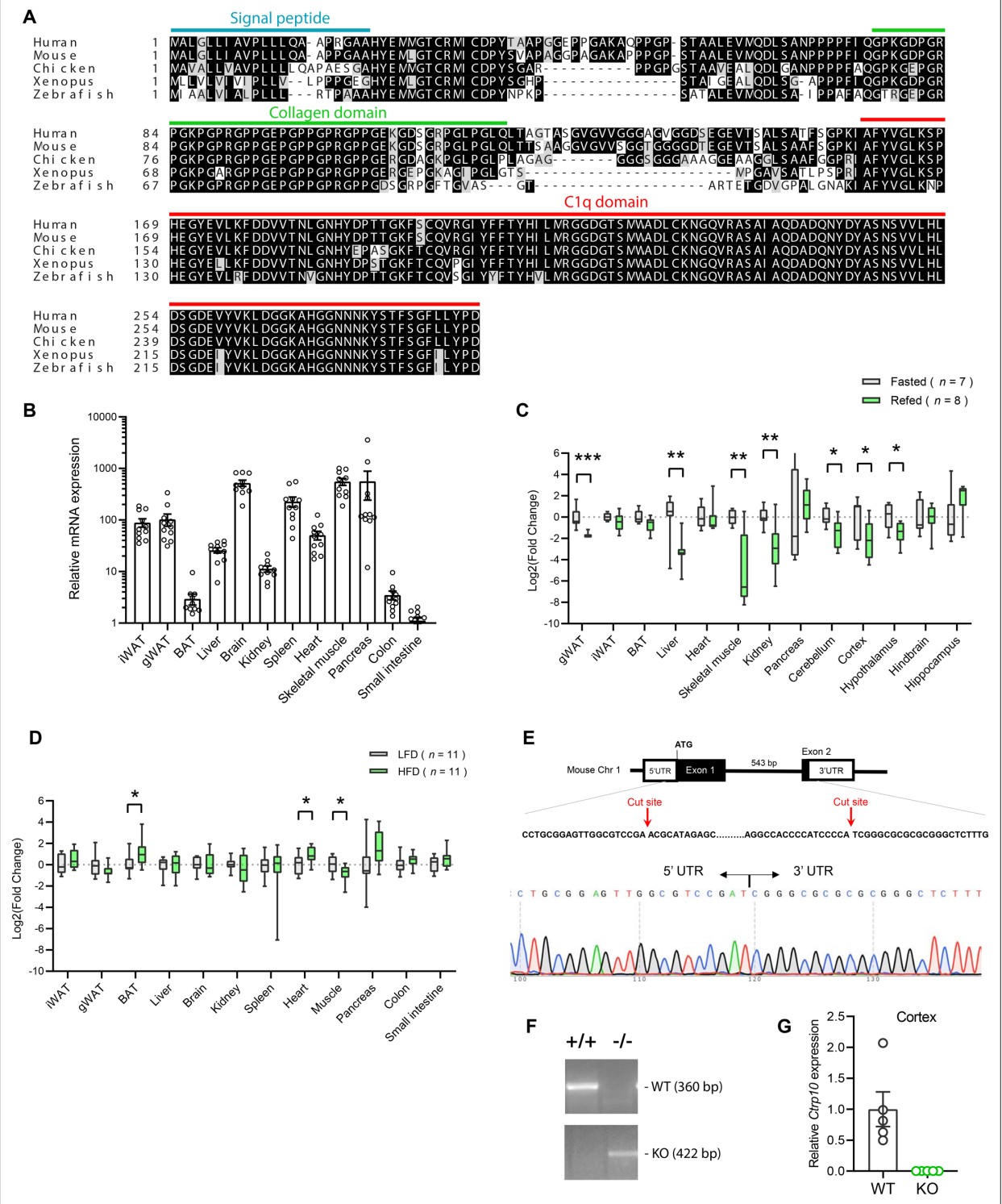

**Figure 1.** Nutritional regulation of *Ctrp10* expression. (**A**) Sequence alignment of full-length human (GenBank # NP_872334), mouse (NP_997116), chicken (XP_046777733), *Xenopus* frog (XP_031749381), and zebrafish (XP_001920705) CTRP10/C1ql2 using Clustal-omega (***Sievers and Higgins, 2018***). Identical amino acids are shaded black and similar amino acids are shaded grey. Gaps are indicated by dash lines. Signal peptide, collagen domain with characteristic Gly-X-Y repeats, and the C-terminal globular C1q domain are indicated. (**B**) *Ctrp10* expression across different mouse tissues (n=10). (**C**) Expression of *Ctrp10* across mouse tissues in response to an overnight (16 hr) fast or fasting followed by 2 hr refeeding. (**D**) Expression of *Ctrp10* across mouse tissues in response to a high-fat diet (HFD) for 12 weeks or a control low-fat diet (LFD). (**E**) Generation of *Ctrp10* knockout (KO) mice. The entire protein coding region in exon 1 and 2 of *Ctrp10* was deleted using the CRISPR/Cas9 method and confirmed with DNA sequencing. (**F**) Wild-type (WT)

*Figure 1 continued on next page*

*Figure 1 continued*

and KO alleles were confirmed by PCR genotyping. (**G**) The complete loss of *Ctrp10* transcript in KO mice was confirmed in mouse cortex, one of the tissues with high *Ctrp10* expression (WT, n=5; KO, n=5). All expression levels were normalized to *β-actin*. All data are presented as mean ± S.E.M. * p<0.05; ** p<0.01; *** p<0.001.

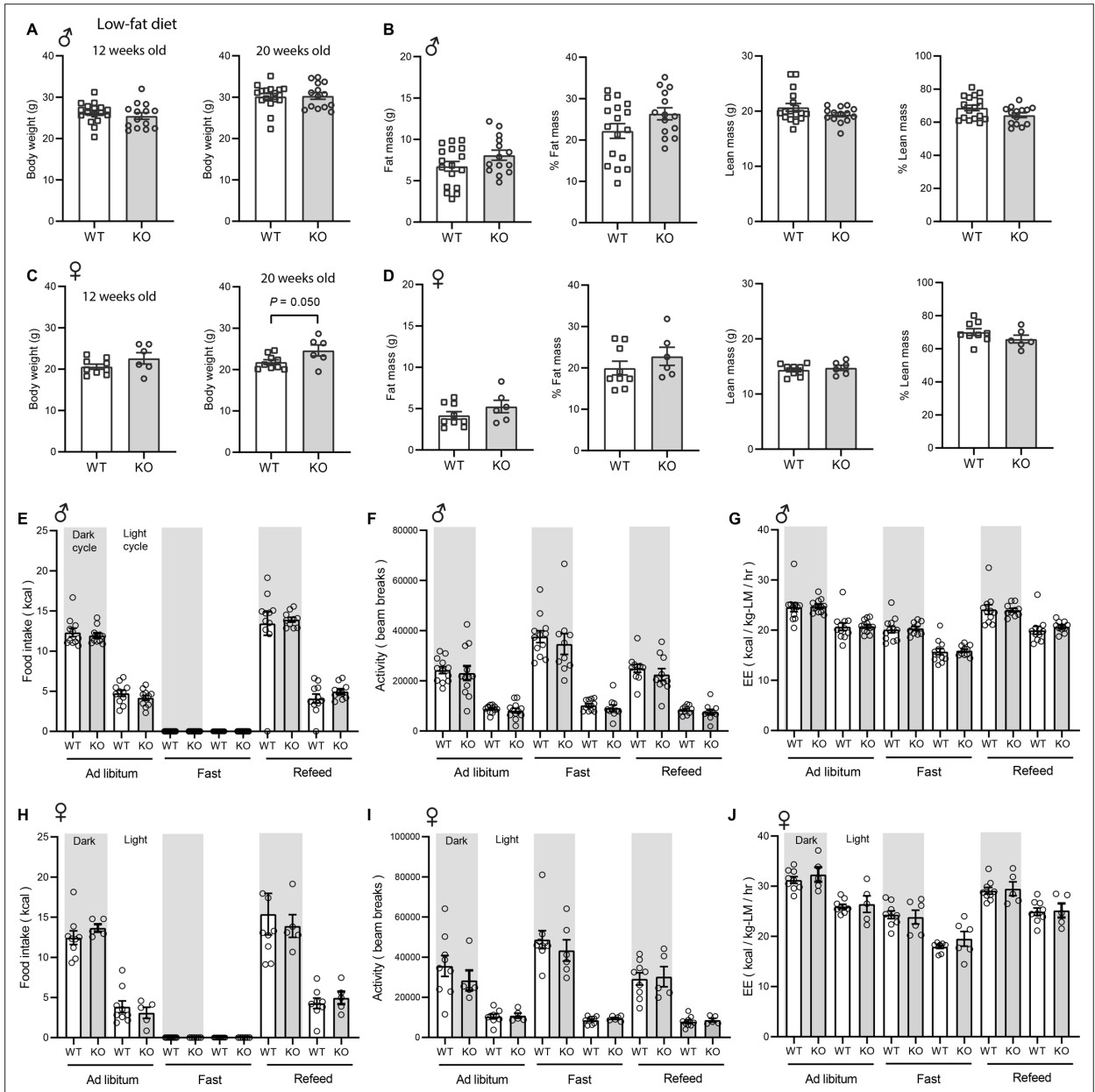

**Figure 2.** *Ctrp10*-KO mice fed a low-fat diet have normal body weight and energy balance. (**A-B**) Body weight (**A**) and body composition analysis (**B**) of fat mass, % fat mass (relative to body weight), lean mass, and % lean mass of WT (n=17) and KO (n=14) male mice at 18 weeks of age. (**C-D**) Body weight (**C**) and body composition analysis (**D**) of fat mass, % fat mass (relative to body weight), lean mass, and % lean mass of WT (n=9) and KO (n=6) female mice at 13 weeks of age. (**E-G**) Food intake, physical activity, and energy expenditure (EE) in male mice at 18 weeks of age across the circadian cycle (light and dark) and metabolic states (ad libitum fed, fasted, refed; WT, n=11–12; KO, n=10–12). (**H-J**) Food intake, physical activity, and energy expenditure in female mice at 13 weeks of age (WT, n=9; KO, n=6). All data are presented as mean ± S.E.M.

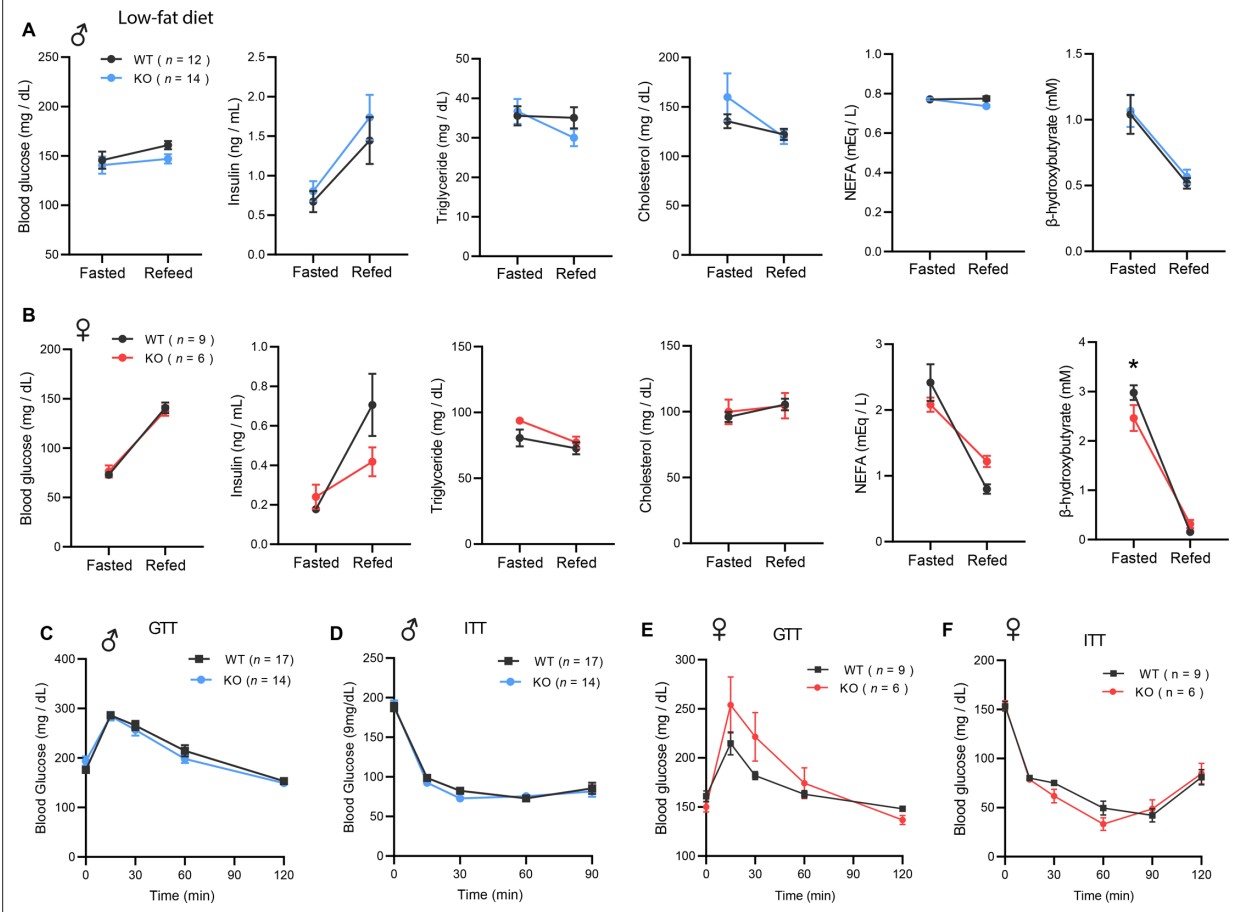

**Figure 3.** *Ctrp10*-KO mice fed a low-fat diet have normal fasting-refeeding response and glucose homeostasis. (**A-B**) Overnight fasted and refed blood glucose, serum insulin, triglyceride, cholesterol, non-esterified free fatty acids (NEFA), and β-hydroxybutyrate levels in male (**A**) and female (**B**) mice. (**C-D**) Blood glucose levels during glucose tolerance tests (GTT; **C**) and insulin tolerance tests (ITT; **D**) in WT (n=17) and KO (n=14) male mice at 12 weeks of age. (**E-F**) Blood glucose levels during glucose tolerance tests (GTT; **E**) and insulin tolerance tests (ITT; **F**) in WT (n=9) and KO (n=6) female mice at 20 and 21 weeks of age, respectively. All data are presented as mean ± S.E.M. * p<0.05 (two-way ANOVA with Sidak's post hoc tests).

data indicate that CTRP10 is dispensable for metabolic homeostasis when mice are young (<20 weeks old) and fed a LFD.

## CTRP10-deficient female mice on a low-fat diet develop obesity with age

Because female KO mice were slightly heavier at 20 weeks of age (***Figure 2C***), we suspected the weight may diverge further with age. Consequently, we monitored the body weight of female mice fed LFD over an extended period. Indeed, the female KO mice gained significantly more weight and adiposity with age (***Figure 4A–C***). By the time the mice reached 40 weeks of age, female KO mice weighed ~6 g (20 %) heavier than the WT controls. Consistent with greater adiposity, the adipocyte cell size (cross-sectional area) was also significantly larger in both gonadal (visceral) white adipose tissue (gWAT) and inguinal (subcutaneous) white adipose tissue (iWAT) (***Figure 4D–E***). Increased weight gain over time was not attributed to differences in food intake, as measured manually over a 24 hr period (***Figure 4F***). Fecal output, frequency, and energy content were also not different between genotypes (***Figure 4G***), suggesting that weight gain was not due to greater nutrient absorption. Deep colon temperatures during both light and dark cycle were also not different between genotypes (***Figure 4H***). Indirectly calorimetry analyses also revealed no significant differences between genotypes in food intake, physical activity, and energy expenditure across the circadian cycle and metabolic states (ad libitum fed, fasted, refed; ***Figure 4I–K***). Despite significantly greater body weight

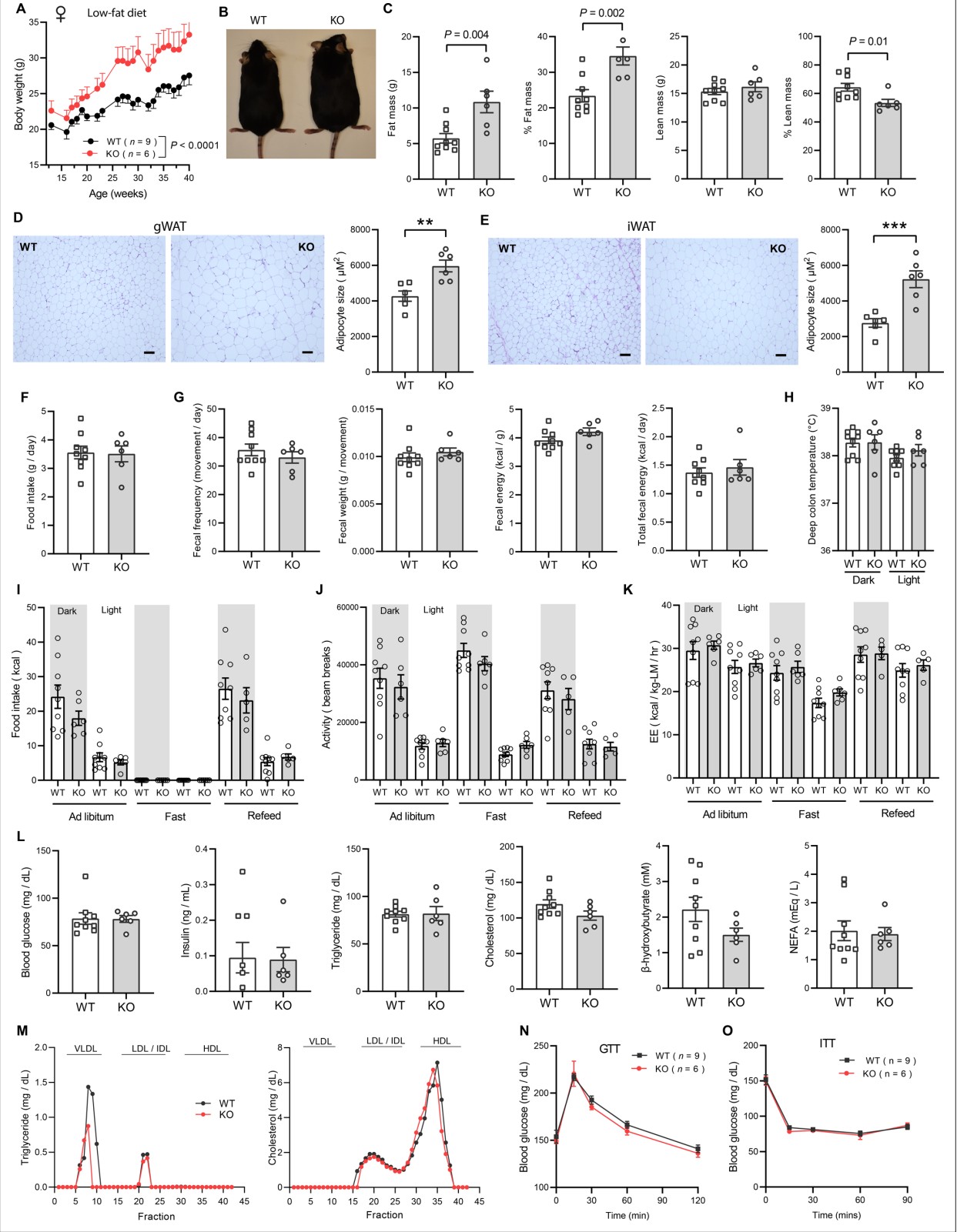

**Figure 4.** *Ctrp10*-KO female mice on a low-fat diet develop obesity with age. (**A**) Body weights over time of WT and KO female mice fed a low-fat diet (LFD). (**B**) Representative image of WT and KO female on LFD for 40 weeks. (**C**) Body composition analysis of WT (n=9) and KO (n=6) female mice fed a LFD. (**D**) Representative H&E stained histology of gonadal white adipose tissue (gWAT) and the quantification of adipocyte cell size (n=6 per genotype). Scale bar = 100 μM. (**E**) Representative H&E stained histology of inguinal white adipose tissue (iWAT) and the quantification of adipocyte cell size (n=6

*Figure 4 continued on next page*

Figure 4 continued

per genotype). Scale bar = 100 μM. (**F**) 24 hr food intake data measured manually. (**G**) Fecal frequency, fecal weight, and fecal energy over a 24 hr period. (**H**) Deep colon temperature measured at the light and dark cycle. (**I-K**) Food intake, physical activity, and energy expenditure in female mice across the circadian cycle (light and dark) and metabolic states (ad libitum fed, fasted, refed; WT, n=9; KO, n=6). Indirect calorimetry analysis was performed after female mice were on LFD for 30 weeks. (**L**) Overnight (16 hr) fasted blood glucose, serum insulin, triglyceride, cholesterol, non-esterified free fatty acids, and β-hydroxybutyrate levels. (**M**) Very-low density lipoprotein-triglyceride (VLDL-TG) and high-density lipoprotein-cholesterol (HDL-cholesterol) analysis by FPLC of pooled (n=6–7 per genotype) mouse sera. (**N**) Blood glucose levels during glucose tolerance tests (GTT). (**O**) Blood glucose levels during insulin tolerance tests (ITT). GTT and ITT were performed when the female mice reached 28 and 29 weeks of age, respectively. WT, n=9; KO, n=6.

The online version of this article includes the following figure supplement(s) for figure 4:

**Figure supplement 1.** No significant differences in serum estradiol levels between WT (n=9) and *Ctrp10* KO (n=6) female mice fed a low-fat diet (LFD).

**Figure supplement 2.** Mitochondrial respiration through complex I (CI), CII, and CIV in liver and skeletal muscle (gastrocnemius) of WT (n=9) and *Ctrp10* KO (n=6) female mice fed a low-fat diet (LFD).

and adiposity, female KO mice had the same metabolic profile as the lean WT controls. Given the important role of sex hormone in systemic metabolism, we measured serum estradiol levels and they were not significantly different between WT and KO female mice (*Figure 4—figure supplement 1*). There were no differences in fasting blood glucose, serum insulin, TG, cholesterol, NEFA, and β-hydroxybutyrate levels between genotypes (*Figure 4L*). Interestingly, VLDL-TG levels were lower in female KO mice whereas HDL-cholesterol level was not different between genotypes (*Figure 4M*). Direct assessments of glucose handling capacity and insulin sensitivity by glucose and insulin tolerance tests, respectively, also revealed no differences between genotypes (*Figure 4N–O*). Measurements of mitochondrial respiration in skeletal muscle (gastrocnemius) showed no significant differences in respiration through complex I (CI), CII, and CIV between genotypes; in contrast, *Ctrp10* KO female mice had higher respiration at CIV relative to WT controls (*Figure 4—figure supplement 2*). Together, these data indicate that *Ctrp10*-KO female mice fed LFD develop obesity, but preserve a largely healthy metabolic profile similar to the much leaner WT female mice.

## Rapid weight gain in CTRP10-deficient female mice fed a high-fat diet

Next, we challenged the mice with a HFD to determine if the sex-dependent effects on body weight become more pronounced. When fed a HFD, body weight gain and body composition were not different between genotypes in male mice (*Figure 5A–B*). Food intake, physical activity, and energy expenditure were also not different between genotypes in male mice across the circadian cycle (light and dark) and metabolic states (ad libitum fed, fasted, refed; *Figure 5C–E*). In striking contrast, female KO mice gained weight rapidly on HFD (~9 g or 28% heavier) and had greater adiposity than the WT controls (*Figure 5F–H*). Weight gain in female mice was not due to differences in food intake, body temperature, or changes in fecal output, frequency, and energy content between genotypes (*Figure 5—figure supplement 1*). We measured serum estradiol levels and they were not significantly different between WT and KO female mice on HFD (*Figure 5—figure supplement 2*). Metabolic cage analysis also revealed no differences in caloric intake, physical activity, and energy expenditure between genotypes in female mice (*Figure 5I–K*). The ANCOVA analysis of energy expenditure using body weight as a covariate also did not reveal any differences between genotypes in female mice (*Figure 5L*). Interestingly, the respiratory quotient (RER) was significantly lower in female KO mice relative to WT controls, especially during fasting and refeeding (*Figure 5M*), suggesting a greater reliance on lipid substrates for energy metabolism during those periods. Interestingly, hepatic, but not skeletal muscle, mitochondrial respiration through CII and CIV was lower in *Ctrp10* KO female mice relative to WT controls on HFD (*Figure 5—figure supplement 3*). Together, these data indicate that CTRP10 is required for female-specific body weight control in response to caloric surplus, but neither food intake, physical activity level, nor whole-body energy expenditure could account for the marked increase in body weight and adiposity.

## Obesity is uncoupled from insulin resistance and dyslipidemia in CTRP10-deficient female mice fed a HFD

To evaluate whether the increased adiposity in KO females led to metabolic dysregulation, we again measured fasting and refeeding responses in WT and *Ctrp10*-KO mice fed a HFD. No differences in fasting and refeeding blood glucose, serum insulin, TG, cholesterol, NEFA, and β-hydroxybutyrate

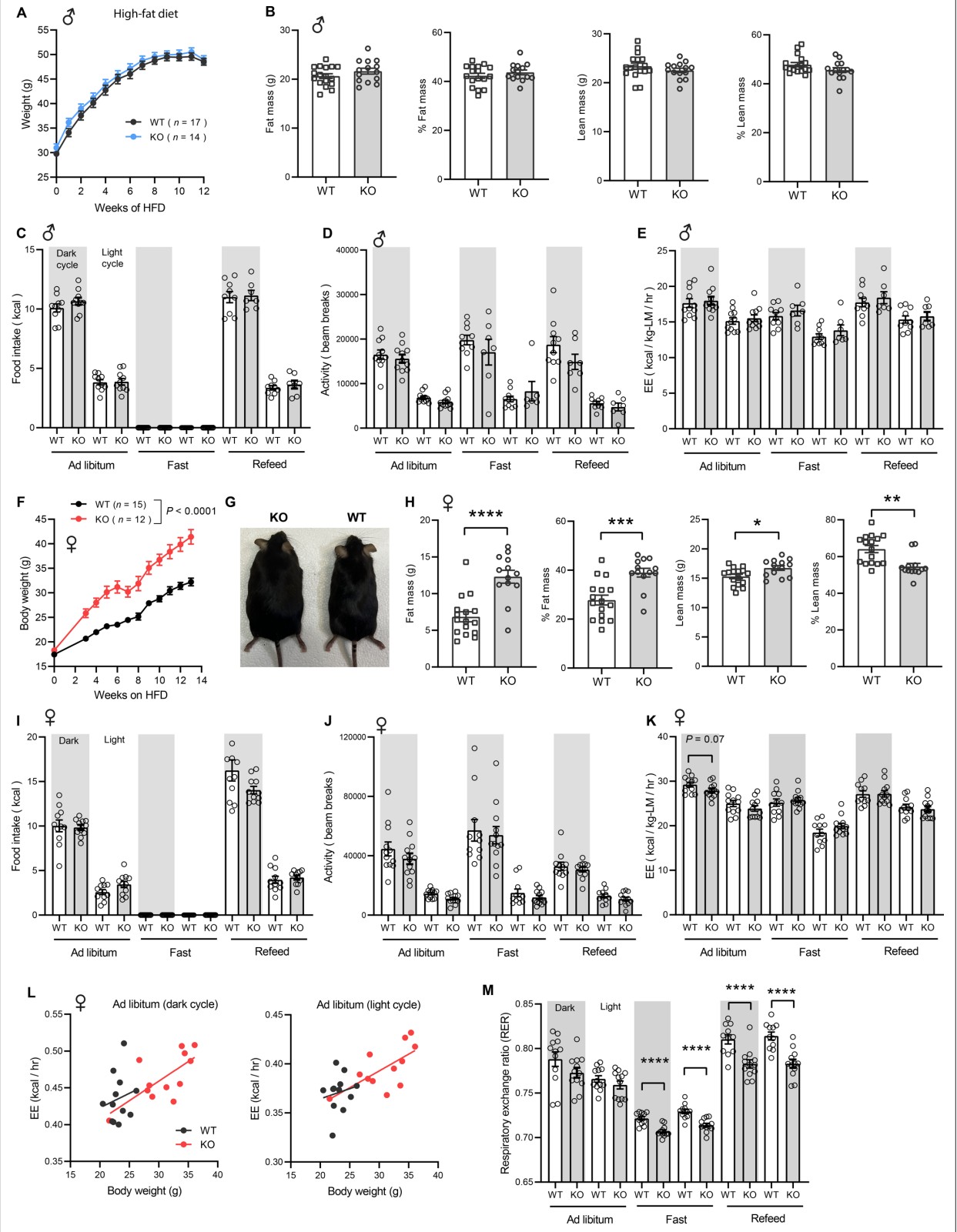

**Figure 5.** Sexually dimorphic response of *Ctrp10*-KO mice to an obesogenic diet. (**A**) Body weights over time of WT and KO male mice fed a high-fat diet (HFD). (**B**) Body composition analysis of WT (n=17) and KO (n=14) male mice fed a HFD for 9 weeks. (**C-E**) Food intake, physical activity, and energy expenditure in male mice across the circadian cycle (light and dark) and metabolic states (ad libitum fed, fasted, refed; WT, n=11; KO, n=11). Indirect calorimetry analysis was performed after male mice were on HFD for 10 weeks. (**F**) Body weights over time of WT and KO female mice fed a high-fat

*Figure 5 continued on next page*

*Figure 5 continued*

diet. (**G**) Representative image of WT and KO female mice after 13 weeks of high-fat feeding. (**H**) Body composition analysis of WT (n=15) and KO (n=12) female mice on HFD for 6 weeks. (**I-K**) Food intake, physical activity, and energy expenditure in female mice (WT, n=11–12; KO, n=12) across the circadian cycle (light and dark) and metabolic states (ad libitum fed, fasted, refed). Indirect calorimetry analysis was performed after female mice were on HFD for 6 weeks. (**L**) ANCOVA analysis of energy expenditure using body weight as a covariate. (**M**) Respiratory exchange ratio (RER). All data are presented as mean ± S.E.M. * $p<0.05$; ** $p<0.01$; *** $p<0.001$; **** $p<0.0001$.

The online version of this article includes the following figure supplement(s) for figure 5:

**Figure supplement 1.** No differences in food intake, body temperature, or changes in fecal output, frequency, and energy content between WT and *Ctrp10* KO female mice fed a high-fat diet.

**Figure supplement 2.** No significant differences in serum estradiol levels between WT (n=6) and *Ctrp10* KO (n=11) female mice fed a high-fat diet (HFD).

**Figure supplement 3.** Mitochondrial respiration through complex I (CI), CII, and CIV in liver and skeletal muscle (gastrocnemius) of WT (n=7) and *Ctrp10* KO (n=8) female mice fed a high-fat diet.

levels were noted between genotypes in male mice (*Figure 6A*). Female KO mice, however, had higher fasting blood glucose and serum insulin levels, and lower β-hydroxybutyrate levels compared to the WT controls (*Figure 6B*). In the refed state, serum insulin levels continued to be significantly higher in female KO mice. Unlike the WT female mice where refeeding markedly lowered serum β-hydroxybutyrate (ketone) levels as expected, KO female mice appeared unable to suppress serum β-hydroxybutyrate levels in response to refeeding (*Figure 6B*).

Consistent with the fasting blood glucose and insulin data, glucose handling capacity and insulin sensitivity assessments by glucose and insulin tolerance in HFD-fed male mice also revealed no differences between genotypes (*Figure 6C–D*). Female KO mice, however, had higher fasting blood glucose and serum insulin levels suggesting the presence of mild insulin resistance (*Figure 6B*). We therefore expected to see differences in either glucose and/or insulin tolerance tests. To our surprise, the rate of glucose clearance in response to glucose or insulin injection was virtually identical between WT and KO female mice (*Figure 6E–F*), suggesting no difference in insulin sensitivity between genotypes. VLDL-TG and HDL-cholesterol profiles were also indistinguishable between WT and KO female mice (*Figure 6G*). Altogether, these data indicate that CTRP10 is not required for metabolic homeostasis in male mice challenged with a HFD. In female mice, however, loss of CTRP10 markedly promotes weight gain in the face of caloric surplus, but, paradoxically, the excess adiposity is largely uncoupled from obesity-linked insulin resistance and dysregulated glucose and lipid metabolism.

## Obesity is uncoupled from adipose dysfunction and hepatic steatosis in *Ctrp10*-KO female mice fed a HFD

Consistent with greater fat mass in visceral (gonadal) fat depot of *Ctrp10*-KO female mice (*Figure 7A*), histological analysis and quantification also indicated significantly larger adipocyte cell size (*Figure 7B*). Although the subcutaneous (inguinal) fat pad weight was also significantly heavier in female KO mice (*Figure 7C*), the adipocyte cell size was marginally bigger but not significant (*Figure 7D*). A bigger fat pad with only marginally larger cell size suggests greater adipocyte hyperplasia in the subcutaneous depot. Increased adipogenesis in response to caloric surfeit is known to be associated with improved systemic metabolic profile (*Vishvanath and Gupta, 2019*).

Obesity is known to be associated with low-grade inflammation (*Hotamisligil, 2006*), fibrosis (*Sun et al., 2013a*), and ER and oxidative stress (*Masschelin et al., 2019*; *Hotamisligil, 2010*). Despite marked differences in body weight and adiposity, the expression of genes associated with inflammation (except for *Ccr2*), fibrosis, oxidative stress in gWAT and iWAT were not significantly different between female KO mice and WT controls (*Figure 7E*). The expression of some genes associated with ER stress (e.g. *Ddit3/CHOP*, *Atf4*, *Xbp1*) in iWAT were actually lower in female KO mice (*Figure 7E*). Corroborating the gene expression data, quantification of hydroxyproline (marker of fibrosis), and malondialdehyde (marker of oxidative stress) revealed no significant differences between genotypes (*Figure 7F–G*).

The liver weight of female KO mice was modestly increased (*Figure 7H*), but when normalized to body weight, it was not significantly different from WT controls (2.76% in WT and 2.60% in KO, p=0.24). Histological analysis and quantification revealed no differences in hepatic lipid content (% lipid area)

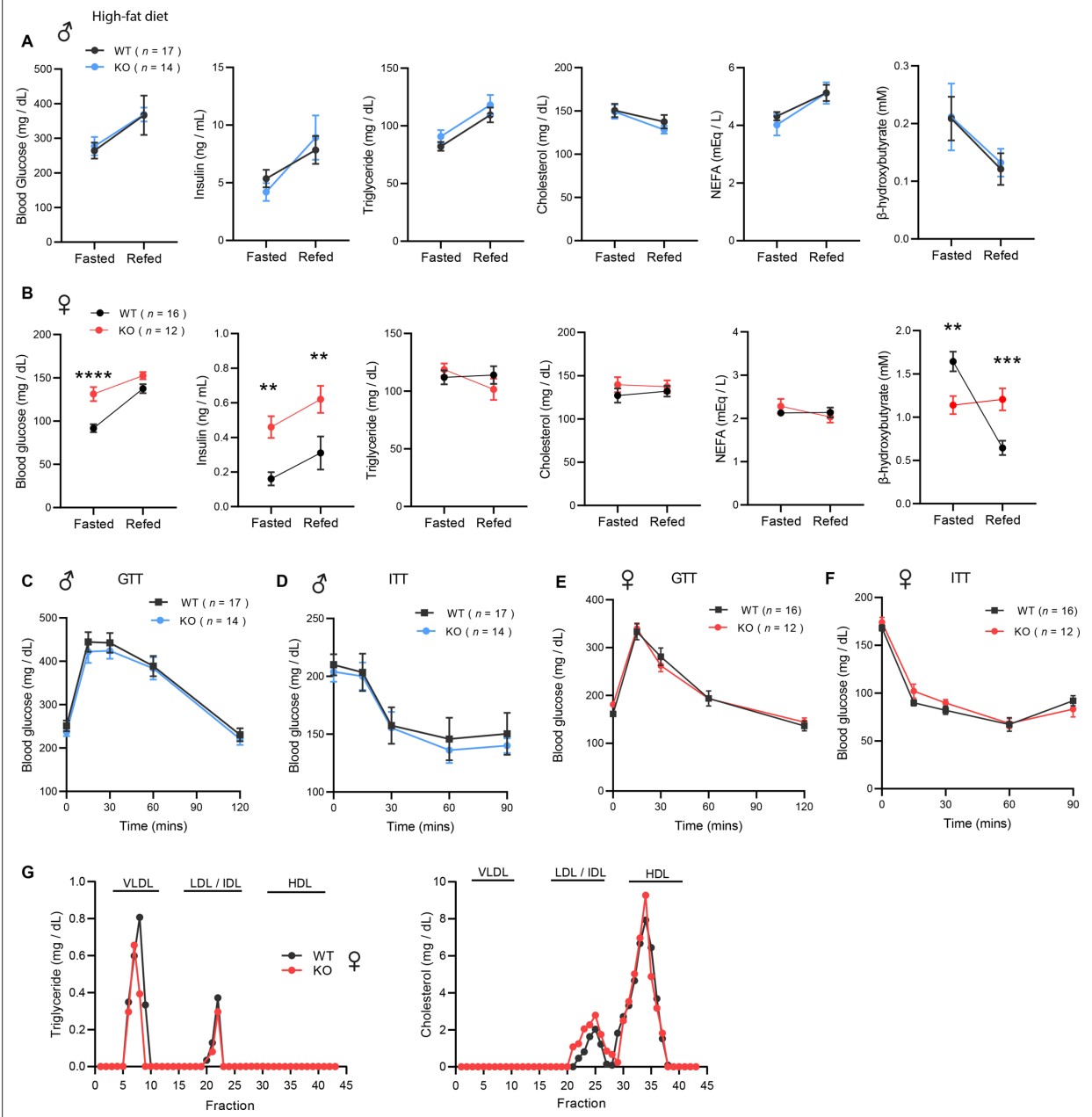

**Figure 6.** *Ctrp10*-KO mice on a high-fat diet (HFD) have normal glucose and insulin tolerance. (**A-B**) Overnight fasted and refed blood glucose, serum insulin, triglyceride, cholesterol, non-esterified free fatty acids (NEFA), and β-hydroxybutyrate levels in male (**A**) and female (**B**) mice fed a HFD for 10 weeks. (**C-D**) Blood glucose levels during glucose tolerance tests (GTT; **C**) and insulin tolerance tests (ITT; **D**) in WT (n=17) and KO (n=14) male mice fed a HFD for 10 weeks. (**E-F**) Blood glucose levels during glucose tolerance tests (GTT; **E**) and insulin tolerance tests (ITT; **F**) in WT (n=16) and KO (n=12) female mice fed a HFD for 8 weeks. (**G**) VLDL-TG and HDL-cholesterol analysis by FPLC of pooled female mouse sera. All data are presented as mean ± S.E.M. ** $p<0.01$; *** $p<0.001$; **** $p<0.0001$ (two-way ANOVA with Sidak's post hoc tests for fasted/refed data).

between genotypes (*Figure 7I*). Interestingly, although hepatic fat content was similar between genotypes, the expression of lipogenic genes (e.g. *Fasn*) was lower and fat catabolism genes (e.g. *Cpt2*, *Ppara*, *Acadl*, *Acadm*, *Acad11*, *Acadvl*) was higher in female KO mice (*Figure 7J*). The expression of genes associated with inflammation, fibrosis, ER and oxidative stress in liver were not significantly different between genotypes (*Figure 7J*). Consistent with the gene expression data, quantification of hydroxyproline (marker of fibrosis) in the liver revealed no significant difference between genotypes (*Figure 7K*). The *Ctrp10* KO female mice, however, had higher levels of malondialdehyde (a marker of oxidative stress) in the liver, suggesting a modest increase in oxidative stress (*Figure 7L*). Altogether,

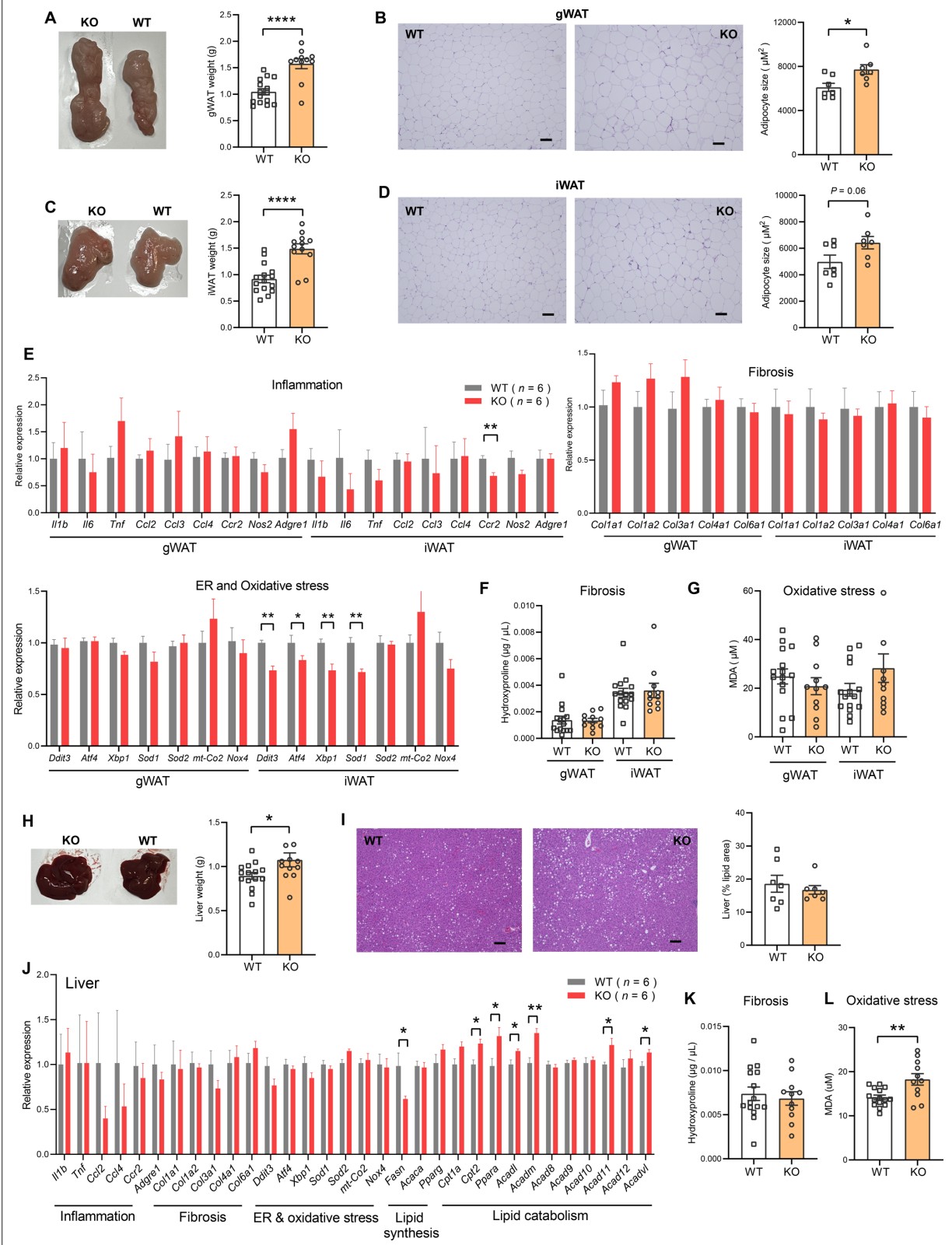

**Figure 7.** *Ctrp10*-KO female mice fed a HFD do not develop adipose tissue dysfunction and fatty liver. (**A**) Representative images of dissected gonadal white adipose tissue (gWAT) and the quantification of gWAT weight in WT (n=15) and KO (n=12) female mice fed a HFD for 14 weeks. (**B**) Representative H&E stained histological sections of gWAT and the quantification of adipocyte cell size (n=7 per genotype). Scale bar = 100 μM. (**C**) Representative images of dissected inguinal white adipose tissue (iWAT) and the quantification of iWAT weight in WT (n=15) and KO (n=12) female

*Figure 7 continued on next page*

Figure 7 continued

mice. (**D**) Representative H&E stained histological sections of iWAT and the quantification of adipocyte cell size (n=7 per genotype). Scale bar = 100 µM. (**E**) Expression of genes associated with inflammation, fibrosis, ER and oxidative stress in gWAT and iWAT of WT (n=6) and KO (n=6) female mice fed a HFD for 14 weeks. Gene expression data were obtained from RNA-seq. (**F-G**) Quantification of hydroxyproline (marker of fibrosis) and malondialdehyde (MDA; marker of oxidative stress) in gWAT and iWAT. WT, n=15; KO, n=11. (**H**) Representative images of dissected liver and the quantification of liver weight in WT (n=15) and KO (n=12) female mice. (**I**) Representative H&E stained histological sections of liver and the quantification of hepatic lipid content (% lipid area; n=7 per genotype). Scale bar = 100 µM. (**J**) Hepatic expression of genes associated with inflammation, fibrosis, ER and oxidative stress, lipid synthesis, and lipid catabolism in WT and KO female mice. Gene expression data were obtained from RNA-seq. (**K-L**) Quantification of hydroxyproline (marker of fibrosis) and malondialdehyde (MDA; marker of oxidative stress) in liver. WT, n=15; KO, n=11. All data are presented as mean ± S.E.M. * p<0.05; ** p<0.01; **** p<0.0001.

these data indicate that obesity is largely uncoupled from inflammation, fibrosis, ER, and oxidative stress in *Ctrp10* KO female mice.

## Transcriptomic and pathway changes associated with the metabolically healthy obesity phenotype in *Ctrp10* KO female mice

To define the specific mechanisms mediating the female-specific effects of *Ctrp10* ablation on favorable metabolic outcomes, four major metabolic tissues (gWAT, iWAT, liver, and skeletal muscle) from female WT and KO mice fed a HFD were subjected to RNA-sequencing (*Figure 8—source data 1–8*). Comparison of differentially expressed genes (DEGs) via limma (*Ritchie et al., 2015*) showed robust changes across tissues, with the largest changes seen in the liver (*Figure 8A–D*). In liver, gWAT, and muscle, we observed comparable numbers of DEGs that were up- and down-regulated, whereas more genes were transcriptionally suppressed in the iWAT of *Ctrp10* KO female mice (*Figure 8E*, top panel). While significant DEGs were identified in all four tissues, only limited overlap was observed between the DEGs in each tissue (*Figure 8E*, bottom panel). Gene set enrichment analyses of the DEGs highlighted distinct and shared processes up- or down-regulated across the four tissues (*Figure 8F*). Pathways and processes related to lipid metabolism and estrogen receptor were the top-ranked up-regulated enrichments across tissues (*Figure 8F*, top panel), whereas processes related to blood clotting and lipoprotein metabolism were the top-ranked down-regulated enrichments (*Figure 8F*, bottom panel).

Of the DEGs, we found significant changes across tissues in relevant classes of genes that encode proteins involved in gene expression (e.g. transcription factors), signaling (e.g. receptors), tissue crosstalk (e.g. secreted proteins), and metabolism (*Figure 9*). Notably, the nuclear receptor, *Nr1d1* (also known as *Rev-Erbα*), is the only gene consistently suppressed across all four tissues (liver, gWAT, iWAT, and muscle) of *Ctrp10* KO female mice (*Figure 8E* lower panel and *Figure 9*). Interestingly, global deletion of *Nr1d1* promotes lipogenesis, adipose tissue expansion, and obesity (*Delezie et al., 2012*; *Hand et al., 2015*). Although the whole-body and adipose-specific *Nr1d1* KO mice fed with HFD become markedly obese, the obesity is not accompanied by insulin resistance, adipose tissue inflammation, and fibrosis (*Hand et al., 2015*; *Hunter et al., 2021*). Like the *Ctrp10* KO female mice, HFD-fed mice lacking NR1D1 can maintain a relatively healthy metabolic profile despite being strikingly obese. Since only male mice were used in these previous studies, we do not know whether female mice lacking NR1D1 would also exhibit similar insulin-sensitive obesity phenotype. In WT mice, NR1D1 acts as a transcriptional repressor of metabolic genes whose expression are upregulated by high-fat feeding; loss of NR1D1 is thought to result in the de-repression of these genes, leading to greater lipid synthesis and fat mass accrual in response to caloric excess (*Hunter et al., 2021*). Thus, the suppression of *Nr1d1* expression—mimicking NR1D1 deficiency—across tissues in HFD-fed *Ctrp10* KO female mice may contribute to benign fat mass expansion without the accompanying adipose tissue fibrosis, inflammation, and oxidative stress.

Among the upregulated genes, *Fgf21* and *Fgf1* were significantly elevated in the liver of *Ctrp10* KO female mice (*Figure 9*). FGF21 is an hepatokine known to improve systemic insulin sensitivity and to promote a favorable metabolic profile in diet-induced obese mice (*Xu et al., 2009*). While hepatic *Fgf21* expression is significantly elevated, its systemic circulating levels were only found to be higher (p=0.051) in LFD-fed, but not HFD-fed, KO female mice (*Figure 9—figure supplement 1*); this suggests liver-derived FGF21 may be retained and act locally within the liver parenchyma. FGF1 is another secreted protein that has been shown to dampen hepatic glucose output by suppressing

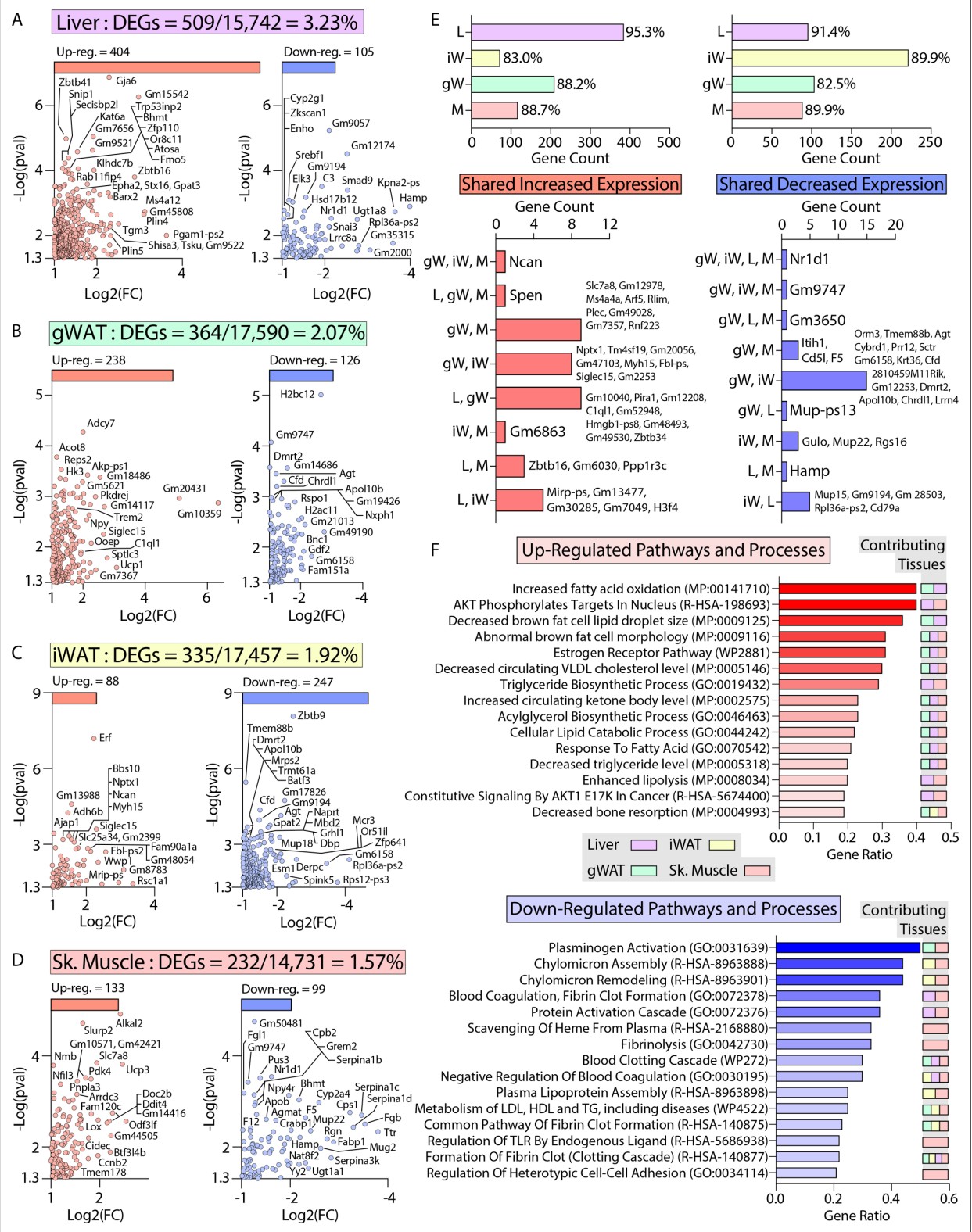

**Figure 8.** Transcriptomic analysis of liver, adipose tissue, and skeletal muscle of female *Ctrp10* KO mice fed a high-fat diet. (**A-D**) Cropped volcano plot views of all differentially expressed genes (DEGs, Log2(Fold Change)>1 or <-1 with a p-value <0.05) of the liver, gonadal white adipose tissue (gWAT), inguinal WAT (iWAT), or skeletal muscle (gastrocnemius). (**E**) Overlap analysis of tissue DEGs showing (top panel) expression unique to gonadal white adipose tissue (gW), inguinal white adipose tissue (iW), liver (L), or skeletal muscle (M). Percent (%) represents percent DEGs unique to each tissue. Bottom panel show DEGs shared across multiple tissues, with all the shared DEGs listed. (**F**) Enrichr analysis (*Kuleshov et al., 2016*) of biological

*Figure 8 continued on next page*

*Figure 8 continued*

pathways and processes significantly (p<0.01) affected across the CTRP10-deficient female mice. Top pathways and processes derived from Gene Ontology (GO), Reactome (R-HAS), WikiPathway human (WP), and mammalian phenotype (MP). All up- or down-regulated DEGs across all tissues were used for analysis. The tissues contributing to the highest ranked pathways and processes are specified. n=6 KO and 6 WT for RNA-seq experiments.

The online version of this article includes the following source data for figure 8:

**Source data 1.** Supplementary table with differentially expressed genes (DEGs) upregulated in the gonadal white adipose tissue (gWAT) of HFD-fed *Ctrp10* KO female mice relative to WT controls.

**Source data 2.** Supplementary table with differentially expressed genes (DEGs) down-regulated in the gonadal white adipose tissue (gWAT) of HFD-fed *Ctrp10* KO female mice relative to WT controls.

**Source data 3.** Supplementary table with differentially expressed genes (DEGs) upregulated in the inguinal white adipose tissue (iWAT) of HFD-fed *Ctrp10* KO female mice relative to WT controls.

**Source data 4.** Supplementary table with differentially expressed genes (DEGs) down-regulated in the inguinal white adipose tissue (iWAT) of HFD-fed *Ctrp10* KO female mice relative to WT controls.

**Source data 5.** Supplementary table with differentially expressed genes (DEGs) upregulated in the liver of HFD-fed *Ctrp10* KO female mice relative to WT controls.

**Source data 6.** Supplementary table with differentially expressed genes (DEGs) down-regulated in the liver of HFD-fed *Ctrp10* KO female mice relative to WT controls.

**Source data 7.** Supplementary table with differentially expressed genes (DEGs) upregulated in the skeletal muscle (gastrocnemius) of HFD-fed *Ctrp10* KO female mice relative to WT controls.

**Source data 8.** Supplementary table with differentially expressed genes (DEGs) down-regulated in the skeletal muscle (gastrocnemius) of HFD-fed *Ctrp10* KO female mice relative to WT controls.

adipose lipolysis (*Sancar et al., 2022*), improve systemic insulin sensitivity by reducing adipose inflammation (*Zhao et al., 2020*), and alleviate hepatic steatosis, inflammation, and insulin resistance (*Fan et al., 2019*). Thus, upregulated expression of *Fgf21* and *Fgf1* in *Ctrp10* KO female mice could contribute to the MHO phenotype. In addition, the upregulated hepatic expression of IL-22 receptor (*Il22ra1*) in *Ctrp10* KO female mice (*Figure 9*) may confer protection against obesity-associated fatty liver, inflammation and fibrosis (*Kong et al., 2012*; *Yang et al., 2010*; *Wang et al., 2014*). Further, a marked increase in uncoupling protein 3 (*Ucp3*) and Krüppel-like factor 15 (*Klf15*) expression in the skeletal muscle (*Figure 9*) may promote lipid utilization and help mitigate lipid-induced insulin resistance in *Ctrp10* KO female mice (*Choi et al., 2007*; *Fan et al., 2021*). Taken together, these combined changes—at the level of gene expression and biological pathways and processes across tissues—acting in concert likely contribute to the apparently healthy obesity phenotype seen in the KO female mice.

## Conservation of mouse DEG co-correlation in humans highlights sex-specific gene connectivity

Next, we asked whether the female-specific transcriptomic effects across tissues were conserved in humans. To address this, we analyzed transcriptional co-correlation of mouse DEG (*Figure 10*) orthologues in GTEx (*GTEx Consortium, 2013*), consisting of 210 males and 100 females filtered for comparison of gene expression across tissues (*Velez et al., 2022*; *Sarver et al., 2023*). Hierarchical clustering of transcriptional correlation of the orthologous DEGs among four metabolic tissues—subcutaneous and visceral white adipose tissue, liver, and skeletal muscle—showed differing patterns of gene connectivity between females (*Figure 10A*) and males (*Figure 10B*). When grouped according to sex in each tissue, the degree of sex-specific gene correlation pairs of DEGs orthologues showed the most significant differences in subcutaneous adipose tissue (*Figure 10C*). Given the whole-body metabolic effects of *Ctrp10* ablation in mice, we further examined the degree of sex-dependent DEG co-correlation across metabolic tissues. This analysis showed that human orthologous genes in subcutaneous adipose tissue (*Figure 10D*, top row) and liver (*Figure 10D*, third row) also exhibited highly significant sex differences in their transcriptional correlation with other DEGs across key metabolic tissues (*Figure 10D*). These analyses highlight the sex-specificity of CTRP10 DEG orthologues in humans, suggest possible sex-biased mechanisms of tissue crosstalk, and overall underscores the conservation of the sex-dependent metabolic function of CTRP10.

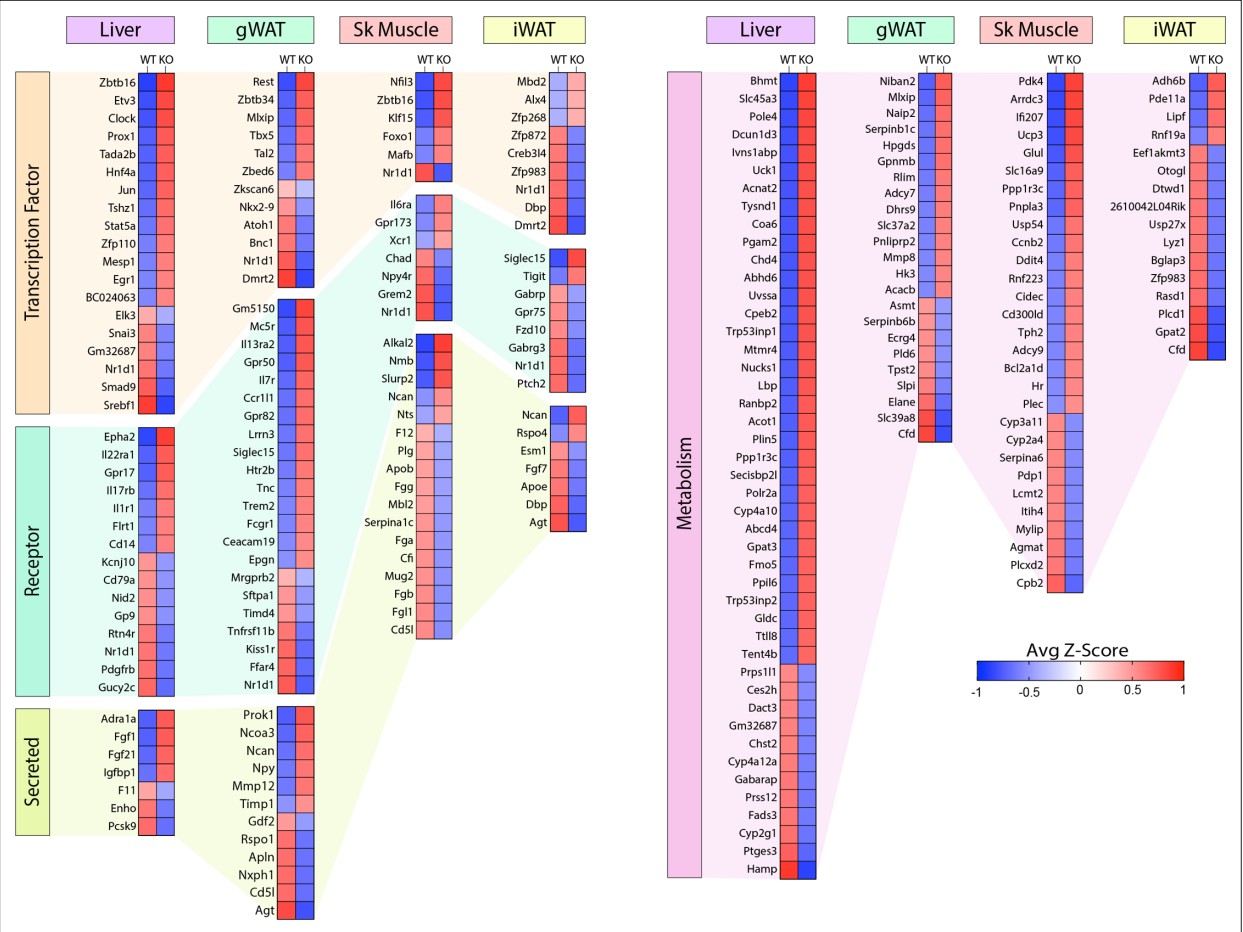

**Figure 9.** Loss of CTRP10 induces significant and wide-spread alterations in the expression of key transcription factors, secreted protein, membrane receptors, and metabolism-associated genes. Selected genes from the DEG list of each tissue organized based on gene type (genes encoding transcription factors, secreted proteins, receptors, and proteins involved in metabolism) and ranked from highest to lowest average row z-score. Z-score is defined as $z = (x-\mu)/\sigma$, where x is the raw score (gene transcript level), $\mu$ is the population mean (i.e., mean of gene expression across both WT and KO samples), and $\sigma$ is the population standard deviation. n=6 per genotype.

The online version of this article includes the following figure supplement(s) for figure 9:

**Figure supplement 1.** While hepatic *Fgf21* expression is significantly elevated, its systemic circulating levels were only found to be higher (p=0.051) in LFD-fed, but no HFD-fed, KO female mice (Figure 9—figure supplement 1).

## Discussion

Our current study has established a novel function for CTRP10 in modulating body weight in a sex-specific manner. When mice were fed a control LFD, female *Ctrp10*-KO mice developed obesity with age; increased adiposity, however, did not impair insulin action and glucose and lipid metabolism. When challenged with an obesogenic diet, female *Ctrp10*-KO mice gained weight rapidly. Despite having strikingly higher adiposity and weighing ~10–11 g (~28%) more, female KO mice fed a HFD exhibited a metabolic profile largely indistinguishable from the much leaner WT controls. Although female KO mice had higher fasting glucose and insulin levels, direct assessments of glucose metabolism and insulin sensitivity by glucose and insulin tolerance tests, however, revealed no differences between genotypes. Except having lower fasting ketone (β-hydroxybutyrate) levels, the fasting lipid profile, as well as VLDL-TG and HDL-cholesterol levels, of *Ctrp10*-KO female mice resembled the WT controls. The hepatic fat content was also comparable between genotypes. Global transcriptomic profiling across different fat depots and liver did not reveal gene expression signatures associated with elevated inflammation, fibrosis, and ER and oxidative stress. Altogether, these findings suggest that CTRP10 deficiency promotes obesity in females but it also uncouples obesity from insulin resistance,

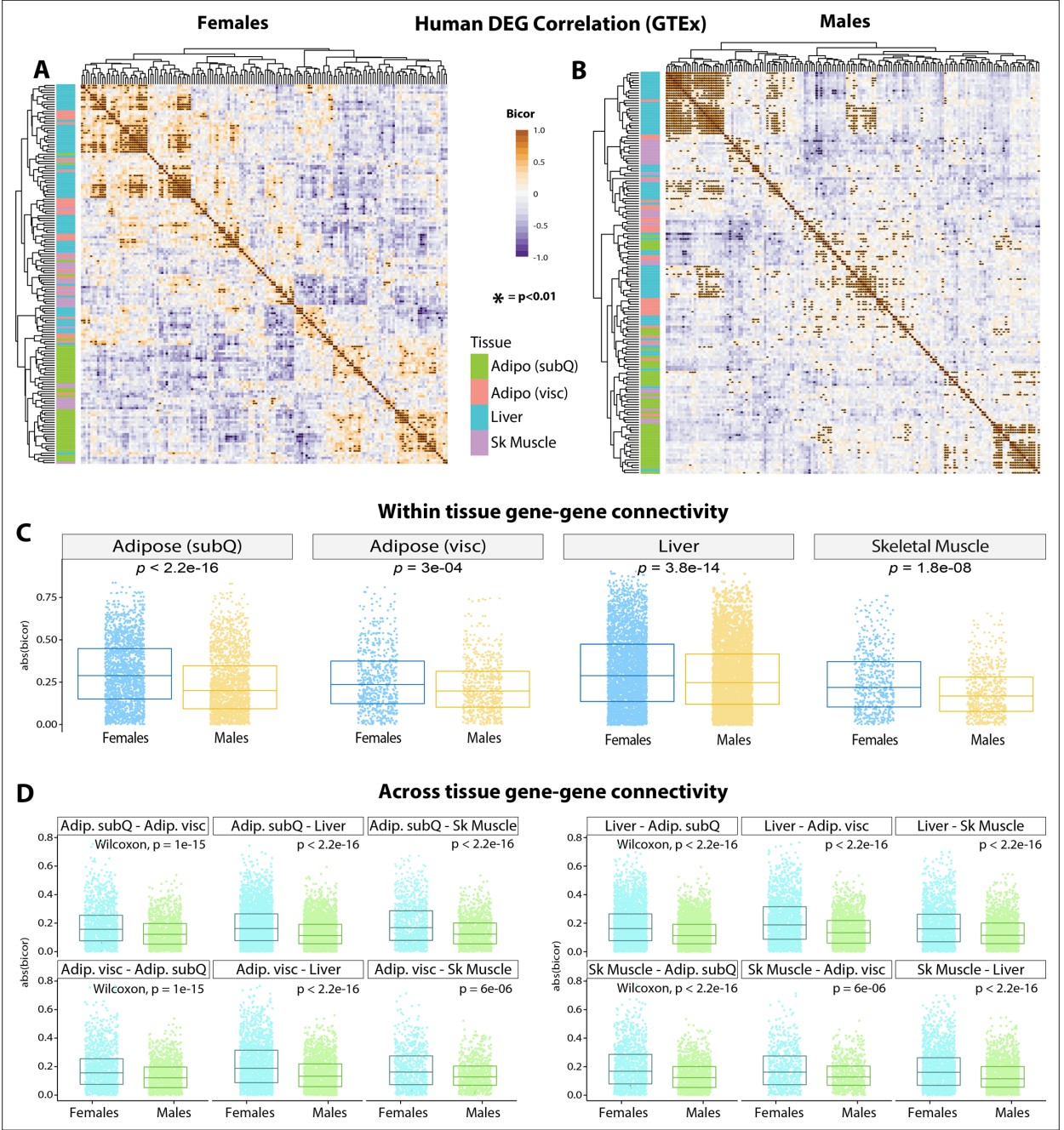

**Figure 10.** GTEx genetic co-correlation of mouse differentially expressed gene (DEG) orthologues. (**A-B**) Heatmaps showing biweight midcorrelation (bicor) coefficient among human tissue DEG orthologs in females (**A**) and males (**B**) in GTEx. Y-axis color indicates tissue of origin, p-value based on students' regression p-value. (**C**) T-tests between correlation coefficient in males and females among all DEG ortholog gene pairs for subcutaneous (SubQ) adipose tissue, visceral (visc) adipose tissue, liver, and skeletal muscle. (**D**) The same as in C, except comparisons are shown for all gene-gene pairs between tissues. For example, the top left graph compared the connectivity of males (blue color) vs females (green color) for correlation between subcutaneous (SubQ) and visceral (Visc) adipose tissue DEG orthologs.

dyslipidemia, steatosis, inflammation, and oxidative stress. Thus, *Ctrp10*-KO female mice represent a novel model of female obesity with largely preserved insulin sensitivity and metabolic health.

Our findings help inform ongoing studies on metabolically healthy obese (MHO) humans (**Klöting et al., 2010**; **Primeau et al., 2011**; **Samocha-Bonet et al., 2012**; **Stefan et al., 2008**; **Calori et al., 2011**; **Aguilar-Salinas et al., 2008**). Because the criteria used to define MHO differs between studies, there is an ongoing debate regarding the prevalence of MHO and what fraction of the MHO population

is insulin-sensitive and metabolically healthy (*Smith et al., 2019*; *Pataky et al., 2010*). Nevertheless, among the obese individuals, there clearly exists a subgroup that maintains long-term normal insulin sensitivity and does not appear to develop any component of the metabolic syndrome (*Smith et al., 2019*). MHO is observed in both sexes, but more common in females (*van Vliet-Ostaptchouk et al., 2014*). The underlying mechanism(s) that uncouple obesity from adverse metabolic health in MHO is not well understood (*Blüher, 2020*; *Samocha-Bonet et al., 2012*; *Smith et al., 2019*). It is currently unknown whether males and females with MHO use similar or distinct mechanism to maintain insulin sensitivity and metabolic health. Our findings in *Ctrp10*-KO female, but not male, mice suggest that there may be female-biased mechanism that prevents metabolic deterioration in the face of obesity, thus underscoring the utility of the *Ctrp10*-KO mice as a female mouse model of MHO.

Obesity is frequently associated with insulin resistance, dyslipidemia, fatty liver, oxidative stress, and chronic low-grade inflammation (*Hotamisligil, 2006*; *Samuel et al., 2010*; *Cohen et al., 2011*). The mechanisms that link obesity to metabolic dysfunctions are complex and multifactorial. There are limited number of mouse models described where obesity is uncoupled from insulin resistance and metabolic health (*Kim et al., 2007*; *Hotamisligil et al., 1996*; *Brandon et al., 2022*; *Kusminski et al., 2012*); in some studies, however, only male mice were used or that the sex of the animals was not specified. In the case of aP2/FABP4 KO male mice, the uncoupling of obesity from insulin resistance was attributed to a marked decrease in TNF-$\alpha$ expression in adipose tissue (*Hotamisligil et al., 1996*). In the case of adiponectin overexpression in leptin-deficient (*ob/ob*) male and female mice, a dramatic expansion of the subcutaneous fat pad is thought to promote lipid sequestration in adipose compartment, thus preventing ectopic lipid deposition in non-adipose tissues (e.g. liver, pancreas, muscle) that would otherwise induce insulin resistance (*Kim et al., 2007*). Massive obesity with preserved insulin sensitivity is also observed in leptin-deficient (*ob/ob*) male and female mice overexpressing the mitochondrial membrane protein, mitoNEET (*Kusminski et al., 2012*). The benign obesity is attributed to the inhibition of iron transport into mitochondria by mitoNEET, leading to reduced mitochondrial activity, fatty acid oxidation, and oxidative stress (*Kusminski et al., 2012*). In the Brd2 hypomorphic mice, severe obesity with lower blood glucose and enhanced glucose tolerance is due to a combination of hyperinsulinemia and marked reduction in macrophage infiltration into fat depot (*Wang et al., 2009*). Lastly, in male mice fed a high-starch diet, the uncoupling of obesity from insulin resistance is associated with lower ceramide levels in liver and skeletal muscle (*Brandon et al., 2022*). In all these cases, the uncoupling of obesity from metabolic dysfunction is seen in either male mice only (female mice were not included) or both sexes. These previous studies suggest that multiple mechanisms, not mutually exclusive, can contribute to the MHO phenotype in different mouse models.

In our study, loss of CTRP10 in female mice largely uncoupled obesity from insulin resistance, dyslipidemia, steatosis, inflammation, and ER and oxidative stress. The preservation of insulin sensitivity in *Ctrp10* KO female mice is due, at least in part, to the absence of obesity-linked adipose and liver inflammation, fibrosis, and oxidative stress. These phenotypes typically associated with a favorable metabolic profile are also observed in MHO individuals (*Primeau et al., 2011*; *Karelis et al., 2005*). A healthy adaptive remodeling of white adipose tissues in response to caloric surfeit helps preserve the storage and secretory function of adipocytes (*Sun et al., 2011*). The expansion of benign adipose tissues further serves to sequester circulating lipids and prevent their ectopic deposition in non-adipose tissue (e.g. liver and skeletal muscle) which can impair insulin action (*Shulman, 2014*; *Virtue and Vidal-Puig, 2010*). The MHO phenotype seen in *Ctrp10* KO female mice reinforce the notion that adipose tissue health, rather than abundance, is an important determinant of metabolic health in obesity.

Because lipidomic analysis was not performed—a limitation of this study—we do not know whether *Ctrp10*-KO female mice have reduced ceramide or diacylglycerol levels in liver and skeletal muscle, two lipid species known to antagonize insulin action (*Samuel and Shulman, 2012*; *Summers et al., 2019*). However, our global transcriptomic and pathway enrichment analysis across visceral and subcutaneous fat depots, liver, and skeletal muscle highlighted the relevant up- and down-regulated pathways and processes (e.g. lipid and lipoprotein metabolism, signaling) that may contribute to the MHO phenotype in *Ctrp10*-KO female mice. How these changes across tissues help to suppress the deleterious effects of obesity and maintain an apparently healthy metabolic profile in *Ctrp10*-KO female mice remains to be fully elucidated. Part of the mechanism may be attributable to the suppression of *Nr1d1* and the upregulated expression of *Fgf1*, *Fgf21*, *Il22ra1*, *Ucp3*, *Klf15*. Increased expression of

these genes is known to reduce obesity-linked inflammation, oxidative stress, steatosis, and insulin resistance. Interestingly, global deletion of *Nr1d1* preserves insulin sensitivity despite promoting lipogenesis, adipose tissue expansion, and obesity (*Delezie et al., 2012*; *Hand et al., 2015*). NR1D1 (also known as REV-ERBα) is a nuclear receptor that acts as a transcriptional repressor (*Zhang et al., 2015*; *Zhang et al., 2016*). Circadian and metabolic genes that normally repressed by NR1D1 are de-repressed in mice lacking NR1D1 globally or in a fat-specific manner, thus leading to fat mass accrual (*Delezie et al., 2012*; *Hunter et al., 2021*). It remains to be determined whether the suppression of *Nr1d1* expression in adipose tissue, liver, and skeletal muscle is indeed responsible, at least in part, for the MHO phenotype seen in *Ctrp10* female mice.

In many single-gene KO mouse models where both sexes are examined, it is often the males that show a more pronounced metabolic phenotype. It is known that C57BL/6 J female mice generally gain significantly less weight on HFD compared to male mice (*Yang et al., 2014*). Therefore, it is intriguing that *Ctrp10*-KO female mice became obese on a control LFD and gained weight rapidly when fed an obesogenic diet. After 12 weeks on HFD, the body weight of female KO mice was approaching that of WT male mice fed the same diet. What mechanism underlies the sexually dimorphic requirement of CTRP10 for body weight control? We know that the obesity phenotype was not attributed to differences in food intake, physical activity level, body temperature, and energy expenditure between WT and KO female mice. We assume that the methods used to quantify these physiologic parameters are sensitive enough to detect small differences that can give rise to divergent body weight over time. Quantification of fecal output and fecal energy content also revealed no differences between genotypes. Thus, loss of CTRP10 did not affect macronutrient intake and absorption. Although we cannot fully rule out the CNS function of CTRP10/C1QL2 (*Matsuda et al., 2016*), our data do not support a central role for CTRP10 in modulating food intake behavior, locomotor activity, and energy expenditure that affect body weight in female mice.

It is known that reduced estrogen level by ovariectomy or blocking estrogen action in estrogen receptor (ERα) KO mice will cause obesity and metabolic dysfunction in female mice fed a HFD (*Rogers et al., 2009*; *Hong et al., 2009*; *Heine et al., 2000*). Conversely, estradiol supplementation decreases HFD-induced weight gain and improves glucose tolerance and insulin sensitivity (*Bryzgalova et al., 2008*; *Stubbins et al., 2012*; *Yonezawa et al., 2012*). Estrogen also has the effect of reducing food intake, and promoting physical activity and energy expenditure (*Mauvais-Jarvis et al., 2013*; *Correa et al., 2015*; *Butera, 2010*; *Xu et al., 2011*; *Krause et al., 2021*). In our study, loss of CTRP10 promotes obesity without altering food intake, physical activity, and energy expenditure. While estrogen's role cannot be completely ruled out, the fact that estrogen levels were not different between genotypes and that female *Ctrp10*-KO mice developed obesity with largely preserved metabolic health suggest that factors other than altered estrogen level contribute to the insulin-sensitive obesity phenotype. Future studies are warranted to uncover what factor(s) is causally contributing to obesity in female mice lacking CTRP10.

The sex-dependent effects of CTRP10 on metabolism and tissue transcriptomes appear to be conserved in humans. When the human orthologues of the mouse DEGs were used to interrogate the GTEx data, clear patterns of gene connectivity within and across metabolic tissues in females and males were observed, with the strongest sex-specific gene correlations seen in subcutaneous adipose tissue and liver. These findings provide further evidence that CTRP10 modulates tissue transcriptome in a sex-dependent manner. Further, our analyses of sex-dependent DEG co-correlation across metabolic tissues also suggest possible sex-biased mechanisms of inter-organ metabolic signaling between adipose tissue and liver.

CTRP10 has been previously shown to bind to the adhesion GPCR, brain angiogenesis inhibitor-3 (Bai3/Adgrb3; *Bolliger et al., 2011*). Bai3 is expressed in the brain and peripheral tissues, and it is a promiscuous GPCR that can bind to multiple ligands. In addition to CTRP10 (C1QL2), Bai3 also binds to CTRP11 (C1QL4), CTRP13 (C1QL3), CTRP14 (C1QL1), neuronal pentraxins, and reticulon 4 (RTN4) receptor (*Hamoud et al., 2018*; *Kakegawa et al., 2015*; *Sigoillot et al., 2015*; *Bolliger et al., 2011*; *Sticco et al., 2021*; *Wang et al., 2021*). A constitutive, whole-body KO of Bai3 mouse models have recently been generated (*Alsharif et al., 2023*; *Shiu et al., 2023*). Both male and female *Bai3* KO mice fed a standard chow have significantly lower body weight, beginning at weaning (3 weeks old) and continue into adulthood (*Alsharif et al., 2023*; *Shiu et al., 2023*). Lower body weight in *Bai3* KO mice of either sex is attributed to a reduction in both lean and fat mass, and is associated with

higher energy expenditure and reduced food intake in male mice (*Alsharif et al., 2023*). The impact of Bai3 deficiency on systemic metabolism in response to a HFD was not examined. The *Ctrp10* KO mice do not phenocopy the phenotypes of the *Bai3* KO mice. When fed a control LFD, the body weight, food intake, and energy expenditure of *Ctrp10* KO male mice were indistinguishable from WT controls. In striking contrast to *Bai3* KO mice, female *Ctrp10* KO mice fed a LFD began to gain more weight around 20 weeks of age, and by 40 weeks had become visibly obese. While we do not rule out CTRP10-Bai3 signaling axis in modulating energy metabolism in peripheral tissues, our findings in *Ctrp10* KO mice suggest that future works are needed to establish the molecular mechanisms that mediate the systemic metabolic function of CTRP10.

Several limitations of our current study are noted. The lack of reliable antibody specific for CTRP10 precludes the assessment of protein abundance in response to altered nutritional and metabolic states. We used a constitutive whole-body KO mouse model of CTRP10 to interrogate its function. It is unknown whether CTRP10 has a role during development that may influence sex-dependent post-natal weight gain with age or in response to a high-caloric diet. Future studies using conditional KO of *Ctrp10* gene in adult mice can help address this issue. We also did not perform tracer studies to address whether there are anabolic and catabolic changes in glucose and fat metabolism in the liver and adipose tissue of *Ctrp10* KO female mice, as this information may shed light on how different tissues handle macronutrients in the absence of CTRP10. In KO female mice, we observed upregulated expression of genes (e.g. *Fgf1*, *Fgf21*, *Il22ra1*, *Ucp3*, and *Klf15*) in liver or skeletal muscle that potentially contributes to the MHO phenotype. The mechanism by which CTRP10 suppresses—either directly or indirectly—the expression of these genes and their de-repression in the absence of CTRP10 is presently unknown. Although the lack of differences in food intake, physical activity, and energy expenditure between genotypes do not support a central role of CTRP10 in mediating the metabolic phenotypes of *Ctrp10*-KO female mice, a brain-specific KO of *Ctrp10* gene is needed to definitively rule this out. Our phenotypic analyses in the context of metabolism are relatively comprehensive but not exhaustive. Although the metabolic profile of obese *Ctrp10*-KO female mice was largely indistinguishable from the much leaner WT controls, some related aspect of metabolic health (e.g. blood pressure and heart function) may be altered in the absence of CTRP10 which we did not examine.

In summary, we have established the physiologic role and requirement of CTRP10 in modulating body weight in a female-specific manner. Importantly, loss of CTRP10 uncouples obesity from insulin resistance and metabolic dysfunction. The CTRP10-deficient female mice represent a unique and valuable model for dissecting female-biased mechanisms that help preserve metabolic health in the face of positive energy balance and increased adiposity.

## Materials and methods
### Mouse models
Eight-week-old mouse tissues (gonadal and inguinal white adipose tissues, interscapular BAT, liver, heart, skeletal muscle, kidney, pancreas, cerebellum, cortex, hippocampus, hindbrain, and hypothalamus) from C57BL/6J male mice (The Jackson Laboratory, Bar Harbor, ME) were collected from fasted and refed experiments as we have previously described (*Rodriguez et al., 2019*). For the fasted group (~10 weeks old), food was removed for 16 hr (beginning 10 hr into the light cycle), and mice were euthanized 2 hr into the light cycle. For the refed group (~10 weeks old), mice were fasted for 16 hr and refed with chow pellets for 2 hr before being euthanized. Tissues (white and brown adipose tissues, liver, whole brain, kidney, spleen, heart, skeletal muscle, pancreas, small intestine, and colon) from C57BL/6 J male mice fed a LFD or a HFD for 12 weeks were also collected as we have previously described (*Rodriguez et al., 2019*).

The *Ctrp10/C1ql2*-KO mice (C57BL/6NCrl-*C1ql2^em1(IMPC)Mbp*/Mmucd; stock number 050587-UCD) were generated using the CRISPR-cas9 method at UC Davis. The two guide RNAs (gRNA) used were 5'-CCGGCGCCGCTCCACCATTACCT-3' and 5'-TCAGGCCACCCCATCCCCATCGG-3'. The *Ctrp10* gene consists of two exons. The entire protein coding region spanning exon 1 and 2 was deleted. This KO strategy ensures a complete null allele for *Ctrp10*. The KO mice were maintained on a C57BL/6J genetic background. Genotyping primers for WT allele were forward (m10-Com-F) 5'-TGTCGGGCTCTTCGACTCTCCA-3' and reverse (m10-WT-R) 5'-GCATCTCGTAGTGAGCCGCTCC-3'. The size of the WT band was 360 bp. Genotyping primers for the *Ctrp10* KO allele were forward (m10-Com-F)

5'-TGTCGGGCTCTTCGACTCTCCA-3' and reverse (m10-Mut-R1) 5'-GTCCAATCAGCTTTCTCAAG TCTGG-3'. The size of the KO band was 422 bp. The genotyping PCR parameters were as follows: 94 °C for 5 min, followed by 10 cycles of (94 °C for 10 s, 65 °C for 15 s, 72 °C for 30 s), then 25 cycles of (94 °C for 10 s, 55 °C for 15 s, 72 °C for 30 s), and lastly 72 °C for 5 min. Due to the presence of GC-rich sequences, 7% DMSO was included in the PCR genotyping reaction. Mice were generated by intercrossing *Ctrp10* heterozygous (+/-) mice, supplemented with intercrossing WT or KO mice. *Ctrp10* KO (-/-) and WT (+/+) controls were housed in polycarbonate cages on a 12 hr light–dark photocycle with ad libitum access to water and food. Mice were fed either a control LFD (10% kcal derived from fat; # D12450B; Research Diets, New Brunswick, NJ) or a HFD (60% kcal derived from fat; #D12492, Research Diets). LFD was provided for the duration of the study, beginning at 5 weeks of age; HFD was provided for 14 weeks, beginning at 6–7 weeks of age. At termination of study, all mice were fasted for 2 hr and euthanized. Tissues were collected, snap-frozen in liquid nitrogen, and kept at –80 °C until analysis. All mouse protocols (protocol # MO22M367) were approved by the Institutional Animal Care and Use Committee of the Johns Hopkins University School of Medicine. All animal experiments were conducted in accordance with the National Institute of Health guidelines and followed the standards established by the Animal Welfare Acts.

## Body composition analysis

Body composition analyses for total fat, lean mass, and water content were determined using a quantitative magnetic resonance instrument (Echo-MRI-100, Echo Medical Systems, Waco, TX) at the Mouse Phenotyping Core facility at Johns Hopkins University School of Medicine.

## Indirect calorimetry

LFD- or HFD-fed WT and *Ctrp10* KO male and female mice were used for simultaneous assessments of daily body weight change, food intake (corrected for spillage), physical activity, and whole-body metabolic profile in an open flow indirect calorimeter (Comprehensive Laboratory Animal Monitoring System, CLAMS; Columbus Instruments, Columbus, OH) as previously described (*Sarver et al., 2020*). In brief, data were collected for 3 days to confirm mice were acclimatized to the calorimetry chambers (indicated by stable body weights, food intakes, and diurnal metabolic patterns), and data were analyzed from the fourth day. Rates of oxygen consumption ($\dot{V}_{O2}$; mL·kg$^{-1}$·h$^{-1}$) and carbon dioxide production ($\dot{V}_{CO2}$; mL·kg$^{-1}$·h$^{-1}$) in each chamber were measured every 24 min throughout the studies. Respiratory exchange ratio (RER = $\dot{V}_{CO2}/\dot{V}_{O2}$) was calculated by CLAMS software (version 5.66) to estimate relative oxidation of carbohydrates (RER = 1.0) versus fats (RER = 0.7), not accounting for protein oxidation. Energy expenditure (EE) was calculated as EE = $\dot{V}_{O2}$× [3.815 + (1.232×RER)] and normalized to lean mass. Because normalizing to lean mass can potentially lead to overestimation of EE, we also performed ANCOVA analysis on EE using body weight as a covariate (*Tschöp et al., 2011*). Physical activities were measured by infrared beam breaks in the metabolic chamber.

## Measurements of 24 hr food intake

To independently confirm the food intake data collected in the metabolic cage (CLAMS), we also performed 24 hr food intake measurements manually. All mice were singly housed, with wire mesh flooring inserts over a piece of cage paper on the bottom of the cage. A known weight of food pellets was given to each mouse. Twenty-four hours later, the leftover food pellets remaining on the flooring insert, along with any spilled crumbs on the cage paper, were collected and weighed. Thus, food intake was corrected for spillage.

## Glucose, insulin, pyruvate, and lipid tolerance tests

All tolerance tests were conducted as previously described (*Lei and Wong, 2019*; *Rodriguez et al., 2016*; *Tan et al., 2016*). For glucose tolerance tests (GTTs), mice were fasted for 6 hr before glucose injection. Glucose (Sigma, St. Louis, MO) was reconstituted in saline (0.9 g NaCl/L), sterile-filtered, and injected intraperitoneally (i.p.) at 1 mg/g body weight (i.e. 10 μL/g body weight). Blood glucose was measured at 0, 15, 30, 60, and 120 min after glucose injection using a glucometer (NovaMax Plus, Billerica, MA). For insulin tolerance tests (ITTs), food was removed 2 hr before insulin injection. 6.5 μL of insulin stock (4 mg/mL; Gibco) was diluted in 10 mL of saline, sterile-filtered, and injected i.p. at

0.75 U/kg body weight (i.e. 10 µL/g body weight). Blood glucose was measured at 0, 15, 30, 60, and 90 min after insulin injection using a glucometer (NovaMax Plus).

## Fasting-refeeding insulin tests

Mice were fasted overnight (~16 hr) then reintroduced to food as described (*Sarver et al., 2022a*). Blood glucose was monitored at the 16 hr fast time point (time = 0 hr refed) and at 2 hr into the refeeding process. Serum was collected at the 16 hr fast and 2 hr refed time points for insulin ELISA, as well as for the quantification of TG, cholesterol, NEFAs, and β-hydroxybutyrate levels.

## Blood and tissue chemistry analysis

Tail vein blood samples were allowed to clot on ice and then centrifuged for 10 min at 10,000 x *g*. Serum samples were stored at –80 °C until analyzed. Serum TGs and cholesterol levels were measured according to manufacturer's instructions using an Infinity kit (Thermo Fisher Scientific, Middletown, VA). NEFAs were measured using a Wako kit (Wako Chemicals, Richmond, VA). Serum β-hydroxybutyrate (ketone) concentrations were measured with a StanBio Liquicolor kit (StanBio Laboratory, Boerne, TX). ELISA kits were used to measure serum insulin (Crystal Chem, Elk Grove Village, IL; cat # 90080), estradiol (Calbiotech, El Cajon, CA; cat # ES380S), and FGF21 (R&D Systems, Minneapolis, MN; cat # MF2100) according to the manufacturer's instructions. Hydroxyproline assay (Sigma Aldrich, MAK008) was used to quantify total collagen content in liver and adipose tissues according to the manufacturer's instructions. Lipid peroxidation levels (marker of oxidative stress) in the liver and adipose tissues were assessed by the quantification of malondialdehyde (MDA) via Thiobarbituric Acid Reactive Substances (TBARS) assay (Cayman Chemical, 700870) according to the manufacturer's instructions.

## Serum lipoprotein triglyceride and cholesterol analysis by FPLC

Food was removed for 2–4 hr (in the light cycle) prior to blood collection. Sera collected from mice were pooled (n=6–7/genotype) and sent to the Mouse Metabolism Core at Baylor College of Medicine for analysis. Serum samples were first fractionated by fast protein liquid chromatography (FPLC). A total of 45 fractions were collected, and TG and cholesterol in each fraction was quantified.

## Histology and quantification

Inguinal (subcutaneous) white adipose tissue (iWAT), gonadal (visceral) white adipose tissue (gWAT), and liver were dissected and fixed in formalin. Paraffin embedding, tissue sectioning, and staining with hematoxylin and eosin were performed at the Pathology Core facility at Johns Hopkins University School of Medicine. Images were captured with a Keyence BZ-X700 All-in-One fluorescence microscope (Keyence Corp., Itasca, IL). Adipocyte (gWAT and iWAT) cross-sectional area (CSA), as well as the total area covered by lipid droplets in hepatocytes were measured on hematoxylin and eosin-stained slides using ImageJ software (*Schneider et al., 2012*). For CSA measurements, all cells in one field of view at ×100 magnification per tissue section per mouse were analyzed. Image capturing and quantifications were carried out blinded to genotype.

## Fecal bomb calorimetry and assessment of fecal parameters

Fecal pellet frequency and average fecal pellet weight were monitored by housing each mouse singly in clean cages with a wire mesh sitting on top of a cutout cardboard that lay at the bottom of the cage for fecal collection. The number of fecal pellets and their total weight was recorded at the end of 24 hr period. Additional fecal pellets collected for 3 full days were combined and shipped to the University of Michigan Animal Phenotyping Core for fecal bomb calorimetry. Briefly, fecal samples were dried overnight at 50 °C prior to weighing and grinding them to powder. Each sample was mixed with wheat flour (90% wheat flour, 10% sample) and formed into 1.0 g pellet, which was then secured into the firing platform and surrounded by 100% oxygen. The bomb was lowered into a water reservoir and ignited to release heat into the surrounding water. These data were used to calculate fecal pellet frequency (bowel movements/day), average fecal pellet weight (g/bowel movement), fecal energy (cal/g feces), and total fecal energy (kcal/day).

## Mitochondrial respirometry of tissue samples

High-resolution respirometry was conducted on previously frozen tissue samples (liver and gastrocnemius muscle) to assay for mitochondrial activity as we have previously described (*Sarver et al., 2024*). Briefly, all tissues were dissected, snap frozen in liquid nitrogen, and stored at –80 °C for later analysis. Samples were thawed in MAS buffer (70 mM sucrose, 220 mM mannitol, 5 mM KH$_2$PO$_4$, 5 mM MgCl$_2$, 1 mM EGTA, 2 mM HEPES pH 7.4), finely minced with scissors, and then homogenized with a glass Dounce homogenizer. The resulting homogenate was spun at 1000 $g$ for 10 min at 4 °C. The supernatant was collected and immediately used for protein quantification by BCA assay (Thermo Fisher Scientific, 23225). Each well of the Seahorse microplate was loaded with 8 μg (liver samples) or 10 μg (gastrocnemius muscle samples) of homogenate protein. Each biological replicate is comprised of three technical replicates. Samples from all tissues were treated separately with NADH (a complex I substrate, 1 mM) or Succinate (a complex II substrate, 5 mM) in the presence of rotenone (a complex I inhibitor, 2 μM), then with the inhibitors rotenone (2 μM) and Antimycin A (4 μM), followed by TMPD (0.45 mM) and Ascorbate (1 mM) to activate complex IV, and finally treated with Azide (40 mM) to assess non-mitochondrial respiration. All respiration data were normalized to mitochondrial content, quantified using MitoTracker Deep Red (MTDR, Thermo Fisher, M22426) as described (*Sarver et al., 2024*). Briefly, lysates were incubated with MTDR (1 μM) for 10 min at 37 °C, then centrifuged at 2000 × $g$ for 5 min at 4 °C. The supernatant was carefully removed and replaced with 1 x MAS solution and fluorescence was read with excitation and emission wavelengths of 625 nm and 670 nm, respectively. To minimize non-specific background signal contribution, control wells were loaded with MTDR and 1 x MAS and subtracted from all sample values.

## Tissue library preparation and RNA sequencing

Total RNA was isolated from tissues using Trizol reagent (Thermo Fisher Scientific) according to the manufacturer's instructions. Library preparation and bulk RNA sequencing of liver, skeletal muscle (gastrocnemius), gonadal white adipose tissue (gWAT), and inguinal white adipose tissue (iWAT) of HFD-fed *Ctrp10*-KO female mice and WT controls were performed by Novogene (Sacramento, California, USA) on an Illumina platform (NovaSeq 6000) and pair-end reads were generated. Sample size: 6 WT and 6 KO for each tissue. All raw sequencing files are available from the NCBI Sequence Read Archive (SRA) accession PRJNA971939.

## Mouse RNA-sequencing analysis

Transcript features were assembled from raw fastq files and aligned to the current version of mouse transcriptome (*Mus musculus*.GRCm39.cdna) using kallisto -aln (*Bray et al., 2016*). Version-specific Ensembl transcript IDs were linked to gene symbols using biomart. Estimated counts were log normalized and filtered for a limit of sum >5 across all samples. Logistic regressions comparing WT vs KO samples across tissues were performed using limma (*Ritchie et al., 2015*). Differential expression results were visualized using available R packages in CRAN: ggplot2, ggVennDiagram and pheatmap. Gene set enrichment analyses of the DEGs were performed using Enrichr (*Kuleshov et al., 2016*). Scripts for analyses and visualization are available at the github link below (Data availability section).

## Human sex difference analysis

All the datasets and scripts to perform analyses are available at the github linked here. Male and Female human data were obtained from Genotype-Tissue Expression (GTEx; *GTEx Consortium, 2013*) and filtered for sufficient comparison of inter-tissue transcript correlation as described (*Velez et al., 2022*; *Sarver et al., 2023*). CTRP10/C1QL2 and other human orthologues for mouse DEGs were identified by intersecting mouse gene symbols with known human orthologues from the vertebrate homology resource at Mouse Genome Informatics (MGI; *The Alliance of Genome Resources Consortium, 2020*). Co-correlation between all human orthologues DEGs were calculated in either self-reported male or female subjects in GTEx using the bicorAndPvalue() function in Weighted Genetic Coexpression Network Analysis (WGCNA) package (*Langfelder and Horvath, 2008*). To compare sex-differences of regression coefficients, Wilcoxon t-tests were compared between coefficients using the R package ggpubr. To investigate the degree of conservation of CTRP-engaged pathways, we mapped the DEGs identified from *Ctrp10* KO versus WT mice to their human orthologs, including human CTRP10/C1QL2, in the GTEx database for transcriptional

correlations. Individuals were stratified by sex to examine sex-specific gene connectivity, consisting of 210 males and 100 females to compare gene expression across tissues. Gene-connectivity analyses were performed based on population correlation significances summarized by cumulative -log10(pvalues) as previously described (*Velez et al., 2022*; *Koplev et al., 2022*; *Massa et al., 2023*; *Zhou et al., 2024*). Additional details on the filtering criteria and normalization methods, as well as the human DEGs file, GTEx data and subject characteristics, can be found in the GitHub repository.

## Quantitative real-time PCR

Total RNA was isolated from tissues using Trizol reagent (Thermo Fisher Scientific). Purified RNA was reverse transcribed using an iScript cDNA Synthesis Kit (Bio-Rad). Real-time quantitative PCR analysis was performed on a CFX Connect Real-Time System (Bio-Rad) using iTaq Universal SYBR Green Supermix (Bio-Rad) according to manufacturer's instructions. Data were normalized to the stable housekeeping gene *β-actin* or *36B4* (encoding the acidic ribosomal phosphoprotein P0) and expressed as relative mRNA levels using the ΔΔCt method (*Schmittgen and Livak, 2008*). Real-time qPCR primers used to assess *Ctrp10* expression across mouse tissues were: *Ctrp10* forward, 5'-cggc ttcatgacacttcctga-3' and reverse, 5'-agcagggatgtgtctttcca-3'. qPCR primers used to confirm the absence of *Ctrp10* in KO mice were: forward (qPCR-m10-F2), 5'-CACGTACCACATTCTCATGCG-3' and reverse (qPCR-m10-R1), 5'-TCGTAATTCTGGTCCGCGTC-3'.

## Statistical analyses

Sample size is indicated in figure and/or figure legend. All results are expressed as mean ± SEM. Statistical analysis was performed with Prism 9 software (GraphPad Software, San Diego, CA). Data were analyzed with two-tailed Student's *t*-tests, one-way ANOVA or two-way ANOVA (with Sidak's post hoc tests). Two-way ANOVA was used for body weight over time, fasting-refeeding response, and all tolerance tests. $p < 0.05$ was considered statistically significant.

## Acknowledgements

This work was supported by the National Institutes of Health (DK084171 to GWW, HL138193 and DK130640 to MMS). DCS is supported by an NIH T32 training grant (HL007534). The fecal bomb calorimetry analysis was performed at the University of Michigan Animal Phenotyping Core, supported by center grants 1U2CDK135066-01 (Mi-MPMOD) and DK020572 (MDRC). The FPLC/serum analyses were performed at the Mouse Metabolism and Phenotyping Core (MMPC) at the Baylor College of Medicine, supported by NIH grants (DK114356 and UM1HG006348).

## Additional information

### Funding

| Funder | Grant reference number | Author |
| --- | --- | --- |
| National Institute of Diabetes and Digestive and Kidney Diseases | DK084171 | G William Wong |
| National Institute of Diabetes and Digestive and Kidney Diseases | DK130640 | Marcus M Seldin |
| National Heart, Lung, and Blood Institute | HL138193 | Marcus M Seldin |
| NIH T32 | HL007534 | Dylan C Sarver |

The funders had no role in study design, data collection and interpretation, or the decision to submit the work for publication.

## Author contributions
Fangluo Chen, Conceptualization, Data curation, Formal analysis, Investigation, Visualization, Writing – review and editing; Dylan C Sarver, Muzna Saqib, Visualization, Conceptualization; Leandro M Velez, Marcus M Seldin, Formal analysis, Methodology, Writing – review and editing, Investigation, Conceptualization; Susan Aja, Formal analysis, Investigation, Methodology, Writing – review and editing; G William Wong, Data curation, Supervision, Funding acquisition, Visualization, Writing – review and editing, Writing – original draft, Project administration

## Author ORCIDs
G William Wong ⬤ https://orcid.org/0000-0002-5286-6506

## Ethics
All mouse protocols (protocol # MO22M367) were approved by the Institutional Animal Care and Use Committee of the Johns Hopkins University School of Medicine. All animal experiments were conducted in accordance with the National Institute of Health guidelines and followed the standards established by the Animal Welfare Acts.

Reviewer #1 (Public review): https://doi.org/10.7554/eLife.93373.3.sa1
Reviewer #2 (Public review): https://doi.org/10.7554/eLife.93373.3.sa2
Reviewer #3 (Public review): https://doi.org/10.7554/eLife.93373.3.sa3
Author response https://doi.org/10.7554/eLife.93373.3.sa4

# Additional files

## Supplementary files
MDAR checklist

## 123451Data availability
All RNA sequencing data have been deposited in the NIH Sequence Read Archive (SRA) under accession PRJNA971939. All processed datasets used and R scripts to reproduce analyses are freely available at GitHub: https://github.com/Leandromvelez/CTRP10-Manuscript-DEG-Sex-specific-connectivities-and-integration/tree/main (copy archived at *Velez, 2024*).

The following dataset was generated:

| Author(s) | Year | Dataset title | Dataset URL | Database and Identifier |
|---|---|---|---|---|
| Chen F, Sarver DC, Saqib M, Velez LM, Aja S, Seldin MM, Wong GW | 2023 | CTRP10 WT and KO pan-tissue Sequencing data in C57BL6/J mice | https://www.ncbi.nlm.nih.gov/bioproject/?term=PRJNA971939 | NCBI BioProject, PRJNA971939 |

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
