## [Editor Report · eLife Assessment]

This manuscript presents a detailed characterization of male and female wildtype and Ctrp10 knockout mice, and reveals that knockout mice develop female-specific obesity that is largely uncoupled from metabolic dysfunction. The data are **convincing**, and the work will be an **important** contribution to understanding how obesity is coupled to metabolic dysfunction, and how this can occur in a sex-specific manner.

---

## [Referee Report · Reviewer #1 (Public review)]

Summary

The manuscript by Chen et al. presents a detailed metabolic characterization of male and female WT and Ctrp10 knockout mice. The main finding is that female KO mice become obese on both low-fat and high-fat diets, but without evidence of marked insulin resistance, hepatic steatosis, dyslipidemia, or increased inflammatory markers. The authors performed a detailed transcriptomic analysis and identified differentially-expressed genes that distinguish high-fat diet -fed Ctrp10 KO from WT control mice. They further show that this set of genes exhibits cross correlation in human tissues, and that this is greater in females than in males. The data indicate that the Ctrp10 KO model may be useful to understand how obesity and metabolic dysfuction are coupled to each other, and how this occurs by a sex-biased mechanism.

Strengths

The work presents a large amount of data, which has been carefully acquired and is convincing. The transcriptomic analysis will further help to define what pathways are associated with obesity, but not necessarily with metabolic dysfunction. The manuscript will be of interest to investigators studying metabolic diseases, and to those studying sex-specific differences in metabolic physiology. The limitations of the study are acknowledged, including that a whole-body knockout was used. The cause of the increased body weight is not entirely clear, despite the careful and detailed analysis that was performed. Notwithstanding these limitations, the phenotype is interesting, and this work will establish basis for further work to understand the mechanisms that are involved.

Weaknesses

The main weaknesses are that no antibody is available to detect Ctrp10, and the knockout is a global knockout since no conditional allele is available. These limitations are discussed in the manuscript. Despite these weaknesses, the current work establishes the intriguing phenotype and its sex-specificity, and will provide a solid foundation for future studies.

---

## [Referee Report · Reviewer #2 (Public review)]

Summary:

Here the authors have shown the role of sex differences in MHO phenotype, which increases the scope for research in this area.

Strengths:

The study provides a detailed idea of how the genes are regulated in sex sex-dependent manner.

Weaknesses:

The mechanistic details are missing

---

## [Referee Report · Reviewer #3 (Public review)]

Summary:

This study examines the impact of CTRP10/C1QL2 absence on obesity and metabolic health in mice. Female mice lacking CTRP10 tend to develop obesity, particularly on a high-fat diet. Surprisingly, they do not display the typical metabolic traits associated with obesity, like fatty liver or glucose intolerance. This indicates a disconnection between weight gain and metabolic issues in these female mice. The research underscores the need to understand sex-specific factors in how obesity influences metabolic health.

Strengths:

The study provides compelling evidence regarding Ctrp10's role in female-specific metabolic regulation in mice, shedding light on its potential significance in metabolically healthy obese (MHO) individuals.

Weaknesses:

-The analysis and description of sex-specific human data require more details to highlight the relevance of Ctrp10 mouse data and the analysis of differentially expressed genes in humans.

-There's a lack of analysis regarding secreted Ctrp10 under various dietary conditions.

---

## [Author Response]

The following is the authors’ response to the original reviews.

**Reviewer #1 (Recommendations For The Authors):**
Although the scripts are available at the github link that is shown, the Readme file is not available as a text file. Spreadsheets summarizing the RNA-seq data ought to be available for download, but these are not present. Likewise, are spreadsheets available for the data used to generate the plots in Fig. 10, so that the identities of particular, correlated genes can be viewed?

We have now included the excel sheet with all the DEGs shown in Figure 8-9 (Figure 8 – Source data 1-8). The source data include DEGs that are up- and down-regulated in gWAT, iWAT, liver, and skeletal muscle. The source data files (excel) are the standard output format. We have also updated the github (https://github.com/Leandromvelez/CTRP10-Manuscript-DEG-Sex-specific-connectivities-and-integration) to include a README file and updated the R scripts to annotate steps and processing considerations. In addition, the README file now contains drive links to the files used the unfiltered kallisto TPM and counts at the transcript-level, as well as resulting Differential Expression results based on genotype. Obviously, all criteria from aligned transcripts such as gene filtering and normalization are included in the scripts provided.

Several items would strengthen the work:(1) Is a CTRP10 antibody available, and does the protein abundance correlate with the mRNA abundances that were assessed in Fig. 1?

Unfortunately, no validated antibody currently exists for CTRP10. Consequently, we were not able to assess protein abundance of CTRP10 in our study.

(2) Were there compensatory changes in the abundance of other CTRP family members? This might be observed at the protein, but not mRNA, level. It might be reasonable to test for the effects of liver, gWAT, skeletal muscle, and iWAT.

We observed no compensatory changes in other CTRP family members based on our RNA-seq data. Unfortunately, we do not have protein data for other CTRP family members.

(3) The gene expression changes shown in Fig. 9 are ranked according to z-score, but it is not clear how this is calculated. It would be helpful to indicate the log2 change in each case.

The z-score is a very commonly used method to show DEGs in studies involving RNA-seq data. We calculate the z-score based on the gene transcript source data (Fig. 8 – Source data 1-8). Z-score is defined as z = (x-μ)/σ, where x is the raw score (gene transcript level), μ is the population mean (mean of gene expression across both WT and KO samples), and σ is the population standard deviation. In essence, the z-score is the raw score minus the population mean, divided by the population standard deviation. We now included this information in Fig. 9 legend.

(4) In Fig. 6, female HFD-fed KO mice had increased glucose (and insulin) after an overnight fast, but increased glucose was not observed in the GTT data. Possibly, this is because the mice were fasted for only 6h for the GTT. This might be mentioned during the description of these data, on lines 221-224. However, this also raises the question of whether there is a difference in the rate of gluconeogenesis (or possibly glycogenolysis for the 6h data) in the KO compared to the controls. Understanding this would require the use of tracers, and is reasonably beyond the scope of this study, but might be mentioned in the discussion.

Per reviewer’s suggestion, we have included this in the “limitation section” of the discussion.

Reduced RER in the HFD-fed female mice might begin to suggest a mechanism since this suggests the mice might have decreased oxidation of carbohydrates and increased oxidation of fat compared to control animals. A glucose tracer might be used to test whether more glucose is stored and, if so, in what tissue this occurs. Possibly, this could be done ex vivo on isolated tissues or cells. Again, this is reasonably beyond the scope of the present study.

Per reviewer’s suggestion, we have included this in the “limitation section” of the discussion.

(5) The discussion includes a brief discussion of the role of estrogen and suggests that in CTRP10 KO mice there are differences in other factors that would be needed to explain the phenotype. Although it is agreed that this is likely the case, estrogen levels were not measured in the present study. It seems like this would be important to study, and might shed light on the female-specific phenotype.

We have now included serum estrogen data. No significant differences in estrogen levels were seen between WT and KO female mice fed either a low-fat diet (Fig. 4 – figure supplement 1) or a high-fat diet (Fig. 5 – figure supplement 2).

**Reviewer #2 (Recommendations For The Authors):**
While the concept is potentially exciting, there are major problems with the current manuscript. It lacks the mechanistic details behind MHO.(1) There is a significant gap that was not addressed by the authors. How exactly does CTRP10 lead to the activation of proteins like Fgf1, Fgf21, Il22ra1, Ucp3, and Klf15 in Ctrp10 knockout female mice? Is it likely that CTRP10 regulates these proteins via indirect mechanisms?

We acknowledge that the lack of mechanistic understanding of how CTRP10 loss-of-function leads to changes in gene expression is a major limitation of the study. We have highlighted this limitation in the discussion section.

• The author notes that Ctrp10 knockout female mice, particularly those on a high-fat diet lack Nr1d1 and can sustain a relatively healthy metabolic state. This is supported by the demonstrated upregulation of Fgf1, Fgf21, Il22ra1, Ucp3, and Klf15 in Ctrp10 knockout female mice. However, the mechanisms through which Ctrp10 knockout influences the expression of these molecules are not elucidated.

We acknowledge that this is a major limitation of the study. We have highlighted this limitation in the discussion section.

• How do you substantiate the role of age and a high-nutrient diet in the development of obesity in knockout female mice? However, it is still unclear whether administering a high-fat diet in >20 week age of mice can develop insulin resistance where obesity is developing in LFD.

When fed a low-fat diet, *Ctrp10*-KO female mice developed obesity with age and yet show little if any glucose intolerance or insulin resistance based on our glucose tolerance and insulin tolerance tests. For the HFD group, we are only comparing WT and KO mice on the same diet (not across diet). While WT mice on HFD gained significant amount of weight over time as expected, *Ctrp10*-KO female mice gain substantially higher amount of weight relative to WT littermates. Despite this, we did not observe a worsening of glucose tolerance and insulin resistance (based on GTT and ITT) in the KO female mice relative to WT controls as we would expect, since greater adiposity in HFD-fed mice generally correlated with worse metabolic outcomes.

(2) The authors should add the NR1D1 dependency study in female mice if possible.

To address would require the generation of Ctrp10/Nr1d1 double KO mouse model and to carry out the entire study again in these double KO mice. Although this suggestion by the reviewer is a good one, this is beyond the scope of the present study.

(3) NR1D1 represses the set of genes that promotes lipogenesis (the author should add some data that validates this statement).

The role of NR1D1 in regulating metabolic genes are extensively documented in the published literature. NR1D1 (also known as REV-ERBα) is a constitutive transcriptional repressor (PMID: 26044300; PMID: 27445394). Many metabolic genes that are normally represses by NR1D1 is de-repressed in mice lacking NR1D1 globally or in the tissue-specific manner (PMID: 26044300; PMID: 34350828; PMID: 22562834). Among the many NR1D1 target genes involved in lipid metabolism include: CD36, Plin2, Elovl5, Acss3 (from: PMID: 26044300); as well as Scd1, Scd2, Pnpla5, Acsl1, Fasn, Hadhb, and Oxsm (from: PMID: 34350828). We have included this information in the discussion section.

(4) The authors should study the effect of Ctrp10 overexpression in HFD-fed female mice and also with KO of CTRP10 in adult mice if possible.

The suggestion by the reviewer is a good one. However, this is beyond the scope of the study. We do not have a *Ctrp10* conditional KO mouse model; as such, we could not study the effect of knocking out CTRP10 in adult mice. Overexpression studies are often considered non-physiological these days since the level of the overexpressed protein is generally much higher than the normal physiological level. For this reason, we did not attempt any overexpression study.

**Reviewer #3 (Recommendations For The Authors):**
Line 114: Could you please provide definitions for "GluK2" and "GluK4" for readers unfamiliar with these terms?

We have now provided definition for these terms.

Line 140: It's stated that skeletal muscle and the pancreas express similar levels of Ctrp10 as the brain. Please double-check and clarify this assertion for accuracy.

In mice, based on our own data (Fig. 1B), *Ctrp10* expression in skeletal muscle and pancreas is comparable to that in the whole brain. In human, based on publicly available data (e.g., Genotype-Tissue Expression portal; GTex), brain expresses much higher level of *CTRP10* transcript relative to other peripheral tissues.

Line 141: Have you investigated whether Ctrp10 levels in plasma change after refeeding? If not, consider exploring this aspect to enhance the comprehensiveness of the study.

No validated antibody currently exists for CTRP10. As such, we could not assess plasma level of CTRP10 after refeeding. We have included this as limitation of our study in the discussion section.

Lines 143-144: Clarify the age bracket of the animals used in the study. Additionally, have you observed similar responses, such as downregulation of Ctrp10 in response to refeeding, in both old and young mice in peripheral tissues?

We have now included the age of the mice (~10 weeks old) for the fasting refeeding study as shown in Fig. 1C in the result and method sections.

Lines 135-149: To complement the experiments shown in Fig 1B-D, provide data pertaining to females.

Ideally, we would like to have this data as well. However, to do this for females would involve 47 mice and the collection of 120 tissues (Fig. 1B; *n* = 10 per tissue), 390 tissues (Fig. 1C; *n* = 7-8 per tissue per fast or refed state), and 528 tissues (Fig. 1D; *n* = 11 per tissue per HFD or LFD). This would be a total of 1038 tissue samples. The main purpose of Fig. 1B-D is to demonstrate that *Ctrp10* transcript is widely expressed and that its expression is modulated by nutritional (HFD vs. LFD) and metabolic (fast vs. refeed) states. These data provided a rationale to examine the metabolic phenotype in mice lacking CTRP10.

To address the reviewer’s point, we looked at the expression levels of CTRP10/C1QL1 between males and females in the Genotype-Tissue Expression (GTEx) database portal and it does not appear that there are sex differences in *CTRP10* expression patterns in normal tissues.

Line 152: Can you provide evidence supporting the hypothesis that Ctrp10 is secreted into the circulation?

CTRP10 has a classic signal peptide sequence and the protein is secreted when expressed in HEK 293 cells (PMID: 18783346). We have shown previously that CTRP10 can be found in the FPLC-fractionated mouse serum using a polyclonal rabbit anti-mouse CTRP10 antibody we generated (PMID: 18783346); this antibody, however, does not work on tissue lysates (many non-specific bands). There is evidence in published literature to show that CTRP10/C1QL2 is clearly found circulating in human plasma. Some of the studies include: (1) Human C1QL2/CTRP10 is detected in the human plasma from UK BioBank (PMID: 37794186; C1QL2 is highlighted in page 335) and serum samples from pregnant females (PMID: 39062451; C1QL2 is highlighted in Table 2). We have included this information in the Introduction section.

Line 178: In Fig 4 D and E (and other figures in the paper), it would be more accurate to express adipocyte size in "μm²" instead of "uM2."

We have double checked and fixed this issue in the figure 4 and 7.

Line 259: Please specify the age of the animals used in the study.

In the method section, we did mention that LFD was provided for the duration of the study, beginning at 5 weeks of age; and that HFD was provided for 14 weeks, beginning at 6-7 weeks of age. Also, in Figure 2A and Figure 4A, the age of the mice is also indicated.

Lines 275-283 and 288-296: It would be more appropriate to move this content to the Discussion section for better contextualization.

We feel that the published information on NR1D1 and FGF21 should be mentioned in the result section so that the readers can immediately appreciate the significance of our data shown in Fig. 8 and 9. However, we also included similar information concerning NR1D1 in the discussion section for better contextualization as suggested.

Line 301: The section on DEG analysis requires additional details. How was the DEG analysis conducted? Were the DEGs from "wild type and KO mice" compared with "human DEGs regulated by sex"? Also, details about the phenotype of the human subjects and their association with obesity should be included. Additionally, discuss specific genes identified by the analysis and their relevance to the Ctrp10 story and human sex-specific gene connectivity analysis.

We have updated the section on DEG analysis and, related to reviewer comments above, significantly expanded the github repository, detailing an analytical walkthrough of all computational analyses performed. To clarify the human integration analysis, we have added the following to the methods:

“To investigate the degree of conservation of CTRP-engaged pathways, we mapped the differentially expressed genes (DEGs) identified from Ctrp10 knockout (KO) versus wild-type (WT) mice to their human orthologs, including human CTRP10, in the GTEx database for transcriptional correlations. Individuals were stratified by sex to examine sex-specific gene connectivity, consisting of 210 males and 100 females to compare gene expression across tissues. Gene-connectivity analyses were performed based on population correlation significances summarized by cumulative -log10(pvalues) as previously described"

Line 330: In Fig 7L, increased oxidative stress in the liver of KO mice is shown. Please provide an explanation for the claim that Ctrp10-KO female mice resembled the WT controls.

In Fig. 7L, we did observe a modest, but significant, increase in oxidative stress in the liver based on the quantification of malondialdehyde (MDA) level, a marker of tissue oxidative stress. However, we did not see any significant differences in the expression of oxidative genes in the liver between WT and KO female mice (Fig. 7J); thus, the statement in line 330 (discussion section) that pertains to oxidative gene expression in fat and liver (Fig. 7E and 7J) is correct.

Line 375: Could you clarify the term "adipose tissue health" and further discuss or provide evidence demonstrating compromised adipose tissue health in female KO mice following HFD?

Adipose tissue health refers to the healthy functioning of adipose tissue (based on its functionality, immune cell population and profile, and metabolic gene expression profiles). Adipose tissue releases free fatty acids in response to fasting and takes up lipids in response to refeeding. Both are these functions are preserved in KO mice as we did not observe any significant differences in free fatty acids (NEFA) and triglyceride levels in the fasted and refed states (Fig. 6AB). Also, we did not observe any significant differences in the expression of inflammatory and fibrotic genes in the adipose tissue of WT and KO female mice fed a high-fat diet (Fig. 7E). If anything, we actually observed a modest, but significant, reduction in the expression of some ER and oxidative stress genes in the KO female mice relative to WT controls (Fig. 7E).

Line 408: Please provide data regarding estrogen levels in wild-type and KO female mice for comparison.

We have now included serum estrogen data. No significant differences in estrogen levels were seen between WT and KO female mice fed either a low-fat diet (Fig. 4 – figure supplement 1) or a high-fat diet (Fig. 5 – figure supplement 2).

Line 587: The GitHub link provided seems to be inactive or incorrect. Please verify and provide the correct link.

We have also updated the github (https://github.com/Leandromvelez/CTRP10-Manuscript-DEG-Sex-specific-connectivities-and-integration) to include a README file and updated the R scripts to annotate steps and processing considerations.

Lines 590-599: Provide additional details about the analysis of human sex-specific genes. Including a table of the top DEGs and pathways differentially regulated by sex would be beneficial for readers' comprehension.

We have expanded the methods, results and associated github repositories to detail all reproducible parameters used in these analyses. The new table of DEGs is included in the manuscript and github repositories.